# Bandwidth Enables Generalization in Quantum Kernel Models

**Abdulkadir Canatar**                                    *acanatar@flatironinstitute.org*
*Center for Computational Neuroscience*
*Flatiron Institute*
*New York, NY 10010, USA*

**Evan Peters**                                              *e6peters@uwaterloo.ca*
*Department of Physics*
*University of Waterloo*
*Waterloo, Ontario, N2L 3G1, Canada*

**Cengiz Pehlevan**                                        *cpehlevan@seas.harvard.edu*
*School of Engineering and Applied Sciences*
*Harvard University*
*Cambridge, MA 0213, USA*

**Stefan M. Wild**                                              *wild@lbl.gov*
*Applied Mathematics & Computational Research Division*
*Lawrence Berkeley National Laboratory*
*Berkeley, CA 94720, USA*

**Ruslan Shaydulin**                              *ruslan.shaydulin@jpmchase.com*
*Global Technology Applied Research*
*JPMorgan Chase*
*New York, NY 10017, USA*

**Reviewed on OpenReview:** *https://openreview.net/forum?id=A1N2qp4yAq*

## Abstract

Quantum computers are known to provide speedups over classical state-of-the-art machine learning methods in some specialized settings. For example, quantum kernel methods have been shown to provide an exponential speedup on a learning version of the discrete logarithm problem. Understanding the generalization of quantum models is essential to realizing similar speedups on problems of practical interest. Recent results demonstrate that generalization is hindered by the exponential size of the quantum feature space. Although these results suggest that quantum models cannot generalize when the number of qubits is large, in this paper we show that these results rely on overly restrictive assumptions. We consider a wider class of models by varying a hyperparameter that we call quantum kernel bandwidth. We analyze the large-qubit limit and provide explicit formulas for the generalization of a quantum model that can be solved in closed form. Specifically, we show that changing the value of the bandwidth can take a model from provably not being able to generalize to any target function to good generalization for well-aligned targets. Our analysis shows how the bandwidth controls the spectrum of the kernel integral operator and thereby the inductive bias of the model. We demonstrate empirically that our theory correctly predicts how varying the bandwidth affects generalization of quantum models on challenging datasets, including those far outside our theoretical assumptions. We discuss the implications of our results for quantum advantage in machine learning.

# 1 Introduction

Quantum computers have the potential to provide computational advantage over their classical counterparts (Nielsen & Chuang, 2011), with machine learning commonly considered one of the most promising application domains. Many approaches to leveraging quantum computers for machine learning problems have been proposed. In this work, we focus on quantum machine learning methods that only assume classical access to the data. Lack of strong assumptions on the data input makes such methods a promising candidate for realizing quantum computational advantage. Specifically, we consider an approach that has gained prominence in recent years wherein a classical data point is embedded into some subspace of the quantum Hilbert space and learning is performed using this embedding. This class of methods includes so-called quantum neural networks (Mitarai et al., 2018; Farhi & Neven, 2018) and quantum kernel methods (Havlíček et al., 2019; Schuld & Killoran, 2019). Quantum neural networks are parameterized quantum circuits that are trained by optimizing the parameters to minimize some loss function. In quantum kernel methods, only the inner products of the embeddings of the data points are evaluated on the quantum computer. The values of these inner products (kernel values) are then used in a model optimized on a classical computer (e.g., support vector machine or kernel ridge regression). The two approaches are deeply connected and can be shown to be equivalent reformulations of each other in many cases (Schuld, 2021). Since the kernel perspective is more amenable to theoretical analysis, in this work we focus only on the subset of models that can be reformulated as kernel methods. A support vector machine (SVM) with a quantum kernel based on Shor's algorithm has been shown to provide exponential (in the problem size) speedup over any classical algorithm for a version of the discrete logarithm problem (Liu et al., 2021), suggesting that a judicious embedding of classical data into the quantum Hilbert space can enable a quantum kernel method to learn functions that would be hard to learn otherwise. Quantum computers are also known to provide quadratic speedup for training a broad class of kernel models Li et al. (2019).

While the quantum kernels provide a much larger class of learnable functions compared to their classical counterpart, the ability of quantum kernels to generalize when the number of qubits is large has been called into question. Informally, Kübler et al. (2021) show that generalization is impossible if the largest eigenvalue of the kernel integral operator is small, and Huang et al. (2021) show that generalization is unlikely if the rank of the kernel matrix is large. The two conditions are connected since for a positive-definite kernel with fixed trace, a small value of the largest eigenvalue implies that the spectrum of the kernel is "flat" with many nonzero eigenvalues. Under the assumptions used by Kübler et al. (2021); Huang et al. (2021), as the number of qubits grows, the largest eigenvalue of the integral operator gets smaller and the spectrum becomes "flat". Therefore, Kübler et al. (2021); Huang et al. (2021) conclude that learning is impossible for models with a large number of qubits unless the amount of training data provided grows exponentially with qubit count. This causes the curse of "exponential" dimensionality (Schölkopf et al., 2002) inherent in quantum kernels. However, Shaydulin & Wild (2021) show that if the class of quantum embeddings is extended by allowing a hyperparameter (denoted "kernel bandwidth") to vary, learning is possible even for high qubit counts. While extensive numerical evidence for the importance of bandwidth is provided, no analytical results are known that explain the mechanism by which bandwidth enables generalization.

In this work, we show analytically that quantum kernel models can generalize even in the limit of large numbers of qubits (and exponentially large feature space). The generalization is enabled by the bandwidth hyperparameter (Schölkopf et al., 2002; Silverman, 2018) which controls the inductive bias of the quantum model. We study the impact of the bandwidth on the spectrum of the kernel using the framework of task-model alignment developed in Canatar et al. (2021), which is based on the replica method of statistical physics (Seung et al., 1992; Dietrich et al., 1999; Mezard & Montanari, 2009; Advani et al., 2013). While nonrigorous, this framework was shown to capture various generalization phenomena accurately compared with the vacuous bounds from statistical learning theory. Together with the spectral biases of the model, task-model alignment quantifies the required amount of samples to learn a task correctly. A "flat" kernel with poor spectral bias implies large sample complexities to learn each mode in a task, while poor task-model alignment implies a large number of modes to learn. On an analytically tractable quantum kernel, we use this framework to show generalization of bandwidth-equipped models in the limit of an infinite number of qubits. Generalization in this infinite-dimensional limit contrasts sharply with previous results suggesting that high dimensionality of quantum Hilbert spaces precludes learning.

Our main contribution is an analysis showing explicitly the impact of quantum kernel bandwidth on the spectrum of the corresponding integral operator and on the generalization of the overall model. On a toy quantum model, we first demonstrate this analytically by deriving closed-form formulas for the spectrum of the integral operator, and show that larger bandwidth leads to larger values of the top eigenvalue and to a less "flat" spectrum. We show that for an aligned target function the kernel can generalize if bandwidth is optimized, whereas if bandwidth is chosen poorly, generalization requires an exponential number of samples on any target. Furthermore, we provide numerical evidence that the same mechanism allows for successful learning for a much broader class of quantum kernels, where analytical derivation of the integral operator spectrum is impossible. While our results do not necessarily imply quantum advantage, the evidence we provide suggests that, even with a compatible, well-aligned task, the quantum machine learning methods require a form of spectral bias to escape the curse of dimensionality and enable generalization.

## 2 Background

We begin by reviewing relevant classical and quantum machine learning concepts and establishing the notation used throughout the paper. We study the problem of regression, where the goal is to learn a target function from data. Specifically, the input is the training set $\mathcal{D} = \{\mathbf{x}^\mu, y^\mu\}_{\mu=1}^P$ containing $P$ observations, with $\mathbf{x}$ drawn from some marginal probability density function $p : \mathcal{X} \to \mathbb{R}$ defined on $\mathcal{X} \subset \mathbb{R}^n$ and $y$ produced by a target function $\bar{f} : \mathcal{X} \to \mathbb{R}$ as $y = \bar{f}(\mathbf{x})$.

**Learning with kernels** Given data in $\mathcal{X}$ distributed according to a probability density function $p : \mathcal{X} \to \mathbb{R}$, we consider a finite-dimensional complex reproducing kernel Hilbert space (RKHS) $\mathcal{H}$ and a corresponding feature map $\psi : \mathcal{X} \to \mathcal{H}$. This feature map gives rise to a kernel function $k(\mathbf{x}, \mathbf{x}') = \langle \psi(\mathbf{x}), \psi(\mathbf{x}') \rangle_\mathcal{H}$. The RKHS $\mathcal{H}$ associated with $k$ is endowed with an inner product $\langle \cdot, \cdot \rangle_\mathcal{H}$ satisfying the reproducing property and comprises functions $f : \mathcal{X} \to \mathbb{R}$ such that $\langle f, f \rangle_\mathcal{H} < \infty$ (Schölkopf et al., 2002). Given a set of $P$ data points $\mathbf{x}^\mu \in \mathcal{X}$, the positive semidefinite Gram matrix is defined elementwise by $\mathbf{K}_{\mu\nu} = k(\mathbf{x}^\mu, \mathbf{x}^\nu)$. The continuous analogue to the Gram matrix $\mathbf{K}$ is the integral kernel operator $T_k : L_2(\mathcal{X}) \to L_2(\mathcal{X})$ defined according to its action:

$$(T_k f)(\mathbf{x}) = \int_\mathcal{X} k(\mathbf{x}, \mathbf{x}') f(\mathbf{x}') p(\mathbf{x}) d\mathbf{x}. \tag{1}$$

By Mercer's theorem (Schölkopf et al., 2002), the eigenfunctions of $T_k$ satisfying $T_k \phi_k = \eta_k \phi_k$ are orthonormal (i.e., $\langle \phi_k, \phi_l \rangle = \delta_{kl}$), span $L_2(\mathcal{X})$, and give rise to an eigendecomposition of $k$ given by $k(\mathbf{x}, \mathbf{x}') = \sum_k \eta_k \phi_k(\mathbf{x}) \phi_k^*(\mathbf{x}')$, where $\{\eta_k\}$ are real-valued, nonnegative eigenvalues of the integral operator due to its Hermiticity. The inner product $\langle \cdot, \cdot \rangle_\mathcal{H}$ in the RKHS of $k$ may be computed with respect to the integral kernel operator of Eq. 1 as $\langle f, g \rangle_\mathcal{H} = \langle f, T_k^{-1} g \rangle_{L_2(\mathcal{X})}$, with the null space of $T_k$ ignored in computing $T_k^{-1}$. From the kernel eigendecomposition, any target $\bar{f} \in L_2(\mathcal{X})$ that lies in the RKHS may therefore be decomposed as $\bar{f}(\mathbf{x}) = \sum_k \bar{a}_k \phi_k(\mathbf{x})$, where $\bar{a}_k = \langle \bar{f}, \phi_k \rangle_\mathcal{H}$ are the target weights. We comment on the case where the target lies outside of the RKHS in Appendix C.

Kernel ridge regression (KRR) is a convex optimization problem over functions that belong to a Hilbert space $\mathcal{H}$ and is stated as follows:

$$f^* = \arg\min_{f \in \mathcal{H}} \frac{1}{2} \sum_{\mu=1}^P \left( f(\mathbf{x}^\mu) - y^\mu \right)^2 + \frac{\lambda}{2} \|f\|_\mathcal{H}^2, \tag{2}$$

where $\|\cdot\|_\mathcal{H}$ denotes the norm with respect to the inner product $\langle \cdot, \cdot \rangle_\mathcal{H}$ defined on the Hilbert space and $\lambda \geq 0$ is the ridge parameter introduced for regularizing the solution.

Using these definitions, one can show that the solution to the regression problem takes the form $f^*(\mathbf{x}) = \mathbf{k}(\mathbf{x})^\top (\mathbf{K} + \lambda \mathbf{I})^{-1} \bar{\mathbf{y}}$, where $\mathbf{k}(\mathbf{x})$ is a vector with elements $\mathbf{k}(\mathbf{x})_\mu = k(\mathbf{x}, \mathbf{x}^\mu)$ and $\bar{\mathbf{y}}_\mu = \bar{y}^\mu$. The kernel trick (Schölkopf et al., 2002) allows one to perform regression without explicitly computing the features if one has access to the analytical kernel function. While providing a rich class of kernels, quantum kernels are typically not expressible analytically and hence require explicit representations of the feature maps.

**Machine learning with quantum computers**   The central motivation for applying quantum computers to problems in machine learning is to leverage the ability of quantum systems to efficiently perform computation in a high-dimensional quantum Hilbert space. We consider quantum systems defined on $n$ qubits whose dynamics may be described using complex-valued linear operators. A general quantum state on $n$ qubits may be described by a positive definite $2^n \times 2^n$ *density matrix* with unit trace and contained in the quantum Hilbert space $\mathcal{H} = \{\rho \mid \rho \succ 0, \mathrm{Tr}(\rho) = 1, \rho \in L(\mathbb{C}^{2^n})\}$, where $L(\mathbb{C}^d)$ denotes bounded linear operators of the form $\mathbb{C}^d \to \mathbb{C}^d$ or, equivalently, $d \times d$ complex matrices. $\mathcal{H}$ is endowed with the inner product $\langle \rho, \rho' \rangle_{\mathcal{H}}$ given by the Hilbert–Schmidt inner product $\langle A, B \rangle_{\mathrm{HS}} = \mathrm{Tr}\{A^\dagger B\}$ for $A, B \in L(\mathbb{C}^d)$.

The dynamics of quantum states are described by applying linear operators to $\rho$, and we here will specifically consider unitary operations (represented by a $2^n \times 2^n$ unitary matrix $U$). Then, the quantum states that we are interested in may be prepared by applying a unitary operator to the vacuum state $|0\rangle$, where the "ket" notation $|j\rangle$ represents the $j$th standard basis vector $\hat{e}_j$ in $\mathbb{R}^{2^n}$. When $n = 1$, quantum states may be represented by the *Bloch sphere*: the constraints $\mathrm{Tr}\{\rho\} = 1$ and $\rho = \rho^\dagger$ mean that the components of a density matrix are parameterized by three real parameters $\mathbf{n} = (n_x, n_y, n_z)$ and the density matrix can be written in terms of Pauli matrices $\vec{\sigma}$ as $\rho = \frac{1}{2}(1 + \mathbf{n} \cdot \vec{\sigma})$. Since $\|\mathbf{n}\| \leq 1$ is required for $\rho \geq 0$ subject to $\mathrm{Tr}\{\rho\} = 1$, we can identify any single-qubit state $\rho$ with a vector $\mathbf{n}$ in the unit sphere $S^2 \subset \mathbb{R}^3$. Similarly, any unitary operation acting on a single qubit may be represented as a sequence of rotations on the Bloch sphere (Nielsen & Chuang, 2011), thus making this representation convenient for visualizing feature maps associated with quantum kernels.

By associating the quantum Hilbert space with a feature space $\mathcal{H}$, we can define a feature map that gives rise to a quantum kernel (Havlíček et al., 2019; Schuld & Killoran, 2019). We consider a data-dependent unitary operator $U(\mathbf{x})$ and prepare a density matrix $\rho(\mathbf{x}) = U(\mathbf{x})|0\rangle\langle 0|U^\dagger(\mathbf{x})$, where $|0\rangle\langle 0|$ represents the $2^n \times 2^n$ matrix with "1" in the top left corner and zeros elsewhere; later we discuss examples of how to construct data-dependent unitary operators. A natural choice for a feature map is then $\psi(\mathbf{x}) = \rho(\mathbf{x})$ with the corresponding inner product $\langle \rho, \rho' \rangle_{\mathcal{H}} = \mathrm{Tr}\{\rho\rho'\}$. Under this feature map, the quantum kernel is defined as $k(\mathbf{x}, \mathbf{x}') = \langle \psi(\mathbf{x}), \psi(\mathbf{x}') \rangle_{\mathcal{H}} = \mathrm{Tr}\{\rho(\mathbf{x})\rho(\mathbf{x}')\}$, which inherits symmetry in its arguments from $\langle \cdot, \cdot \rangle_{\mathcal{H}}$. With this association between the feature map $\psi(\mathbf{x})$ and the quantum state $\rho(\mathbf{x})$, we can freely apply existing theory for kernel methods in terms of complex vector spaces to study the spectra and generalization behavior of quantum kernels. In Appendix A we provide further details on the theory of quantum states and kernels, such as construction of the RKHS from Hermitian linear operators (or *observables*) and the geometry and probabilistic nature of quantum operations. In practice, a quantum kernel method computes the kernel matrix entries on a quantum computer by evaluating the value of the observable $|0\rangle\langle 0|$ on the state $U(\mathbf{x})U(\mathbf{x}')|0\rangle\langle 0|U^\dagger(\mathbf{x}')U^\dagger(\mathbf{x})$.

## 3   Motivating example: no generalization without bandwidth

Despite existing techniques for empirically evaluating the potential performance of quantum kernels on classical datasets (Huang et al., 2021) and examples of successful implementation on currently available quantum computers (Glick et al., 2021; Peters et al., 2021; Wu et al., 2021; Hubregtsen et al., 2021), it is often still unclear how to construct a quantum kernel that will be suitable for learning on a real-world classical dataset. This uncertainty in the potential performance of quantum machine learning methods is compounded by regimes in which generalization is apparently impossible with a subexponential amount of training data. These regimes arise from the same feature of quantum computing that originally motivated quantum kernel methods: the availability of exponentially large feature spaces. In this section we discuss a simple example where the high dimensionality of the feature space precludes learning with fixed quantum embeddings, and we show that the introduction of bandwidth enables generalization.

One example of how high dimensionality of the feature space precludes learning is provided by random feature maps. Given a feature map $\psi$ consisting entirely of independent and identically distributed Gaussian features, the operator $\Sigma = \mathbb{E}_{\mathcal{X}}[\psi\psi^\dagger]$ is proportional to the identity operator. Since $\Sigma$ shares eigenvalues with $T_k$ of Eq. 1 (Appendix A), the eigenvalues of $\mathbf{K}$ concentrate around unity at a rate $\mathcal{O}(P^{-1/2})$ (Rosasco et al., 2010). Analogously, we consider a quantum feature map where states $\rho(\mathbf{x})$ are prepared by $2^n$-dimensional random unitaries $U(\mathbf{x})$, with the uniform distribution over unitaries being described by the *Haar measure* (e.g., Collins & Nechita (2016)). Then, identifying a correspondence $\Sigma \to \mathbb{E}_{U \sim U(2^n)}[\rho(\mathbf{x}) \otimes \rho^T(\mathbf{x})]$ in the

quantum case and applying standard results from measure theory, one can directly compute the spectrum of $\Sigma$ (Appendix A). This computation yields $2^{n-1}(2^n + 1)$ nonzero eigenvalues with magnitude $2^{1-n}(2^n + 1)^{-1}$, and thus the nonzero eigenvalues of $\mathbf{K}$ again become uniform as $n \to \infty$. Since the largest eigenvalue is exponentially small in $n$, generalization requires the number of data points to be exponentially large in $n$. The connection between the magnitude of the largest eigenvalue and the required number of training samples can be seen directly from Kübler et al. (2021, Theorem 3), although it is a straightforward consequence of many older results, for example, (Dietrich et al., 1999; Bengio et al., 2005; Liang et al., 2019; Bordelon et al., 2020; Canatar et al., 2021). In other words, efficient generalization becomes impossible when the quantum feature map uses the full extent of the quantum state space uniformly. Our results demonstrate that the converse is true: restricting embeddings to a smaller region of quantum state space recovers the possibility of efficient learning.

### 3.1 Limitations of fixed quantum embeddings

To make the example concrete, we consider the following learning problem. This learning problem and the failure of fixed quantum embeddings on it were considered in Huang et al. (2021, Supplementary Information 9). For an input $\mathbf{x} \in \{0, \pi\}^n$, the goal is to learn a target function $\bar{f}(\mathbf{x}) = \cos\left(x^{(n)}\right)$, where $x^{(n)}$ is the last element of the vector $\mathbf{x}$. Since learning this function is equivalent to learning the value of the last element of $\mathbf{x}$, it is trivial classically, and a simple linear regression succeeds. We now show how KRR with a badly designed quantum kernel fails on this trivial task, and we show how the introduction of bandwidth allows the KRR with a quantum kernel to learn the target.

Consider a quantum kernel equipped with feature map

$$U(\mathbf{x}) = \bigotimes_{j=1}^n U(x^{(j)}), \qquad U(x^{(j)}) = R_x\left(x^{(j)}\right) = \begin{pmatrix} \cos\left(x^{(j)}/2\right) & i\sin\left(x^{(j)}/2\right) \\ i\sin\left(x^{(j)}/2\right) & \cos\left(x^{(j)}/2\right) \end{pmatrix}, \tag{3}$$

where $\bigotimes_{j=1}^n$ is the tensor product of single-qubit operations and $R_x(\theta)$ represents a rotation of $\theta$ about the $x$-axis of the single-qubit Bloch sphere (see Fig. 1A). For this feature map, the embedding factorizes over qubits as $\rho(\mathbf{x}) = U(\mathbf{x})|0\rangle\langle 0|U^\dagger(\mathbf{x}) = \bigotimes_{j=1}^n \rho(x^{(j)})$, and

$$\rho(x^{(j)}) = \begin{pmatrix} \cos^2(x^{(j)}/2) & -i\cos\left(x^{(j)}/2\right)\sin\left(x^{(j)}/2\right) \\ i\cos\left(x^{(j)}/2\right)\sin\left(x^{(j)}/2\right) & \sin^2(x^{(j)}/2) \end{pmatrix}, \tag{4}$$

with the kernel given by

$$k(\mathbf{x}, \mathbf{x}') = \mathrm{Tr}\left(\rho(\mathbf{x})\rho(\mathbf{x}')\right) = \prod_{j=1}^n \cos^2\left(\left(x^{(j)} - x'^{(j)}\right)/2\right). \tag{5}$$

While this kernel is obtained via quantum operations, the fact that it has a closed form expression makes it classically easy to simulate. Nevertheless, it is a useful toy model (Kübler et al., 2021; Huang et al., 2021) for the analytical analysis of exponentially large quantum feature spaces. Since the input data is $\mathbf{x} \in \{0, \pi\}^n$, the kernel becomes a delta function: $k(\mathbf{x}, \mathbf{x}') = \delta_{\mathbf{x}, \mathbf{x}'}$. Therefore, any two points in the feature space $\rho(\mathbf{x})$ and $\rho(\mathbf{x}')$ for $\mathbf{x} \neq \mathbf{x}'$ are orthogonal and the kernel cannot capture the correlations in data. KRR with this kernel simply memorizes the training set and cannot generalize to any target function with a subexponential (in $n$) training set size.

### 3.2 Bandwidth enables learning

The preceding example highlights how quantum kernel methods utilizing high-dimensional spaces can fail to generalize. Our central technique for mitigating this limitation will be to introduce bandwidth to quantum kernels (Shaydulin & Wild, 2021). We now reconsider the kernel with the feature map of Eq. 3 and introduce

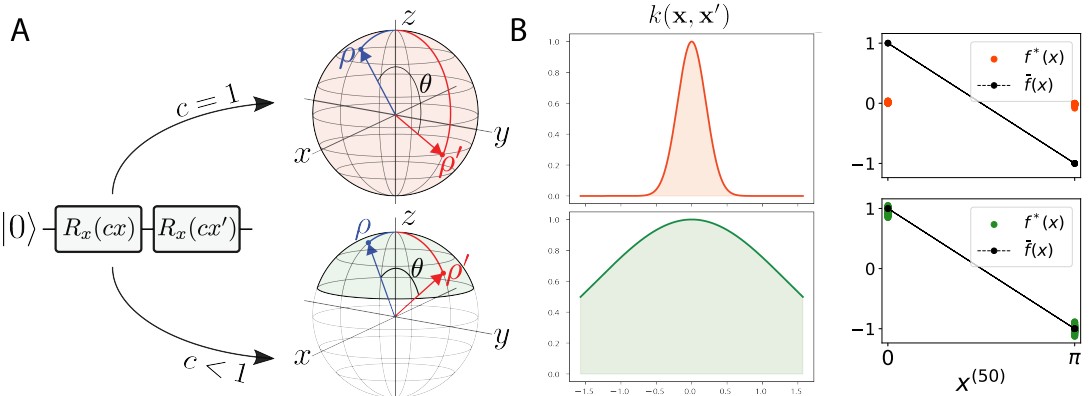

Figure 1: **A** A quantum kernel method involves embedding data using a quantum circuit, often involving rotation by some *angle* about an axis. When angles are not rescaled properly, data can be embedded far apart in a space with dimensionality $O(2^n)$ (Sec. 3). Similarly, $\lambda_{\max}$ is suppressed as the mean embedding $\mathbb{E}_{\mathcal{X}}[\rho(\mathbf{x})]$ approaches the center of the Bloch sphere (Kübler et al., 2021). **B** High-dimensional feature space results in $k(\mathbf{x}, \mathbf{x}')$ with narrow width (Eq. 6 with $n = 50$ and (top) $c = 1$ (bottom) $c = 0.25$) Tuning bandwidth escapes the "curse of dimensionality" associated with high-dimensional feature space. For $n = 50$ and $\bar{f}(\mathbf{x}) = \cos\left(x^{(50)}\right)$, quantum features that are nearly orthogonal result in a *narrow* kernel ($c = 1$, top) and failure to generalize, while tuning bandwidth ($c < 1$, bottom) recovers KRR generalization performance.

a scaling parameter $c \in [0, 1]$ that controls the bandwidth of the kernel. The feature map and the kernel become

$$U(x^{(j)}) = R_x\left(cx^{(j)}\right), \qquad k(\mathbf{x}, \mathbf{x}') = \prod_{j=1}^{n} \cos^2\left(c\left(x^{(j)} - x'^{(j)}\right)/2\right). \tag{6}$$

Geometrically, the factor $c$ restricts features $\rho(\mathbf{x})$ to a smaller region of the Bloch sphere (Fig. 1A). Consequently, the kernel matrix is no longer diagonal, and we can straightforwardly check that simply tuning $c$ allows KRR with kernel Eq. 6 to generalize (see Fig. 1B).

## 4    Effect of bandwidth on scaling and spectra

In the preceding section we provided a qualitative mechanism for how bandwidth improves the generalization of quantum kernels. We now analyze the expected generalization error of quantum kernels equipped with bandwidth. We derive explicitly the spectrum of the bandwidth-equipped quantum kernel with the feature map Eq. 6 and show how the bandwidth makes the spectrum less "flat," thereby enabling learning. Our main tool is the theory developed in Bordelon et al. (2020); Canatar et al. (2021) where the generalization error, as a function of the training set size, is analytically obtained from the eigenvalues of the kernel and the projection of the target function on the RKHS defined by the kernel.

### 4.1    Explicit formulas for generalization of bandwidth-equipped kernels

We consider the quantum kernel with the feature map given by Eq. 6 and the input distribution $\mathbf{x} \sim \mathrm{Unif}([-\pi, \pi]^n)$, as previously studied by Kübler et al. (2021). For a single qubit, the kernel becomes $k(x, x') = \cos^2(c(x - x')/2)$ with bandwidth parameter $c$, and the eigenvalues can be computed as follows (see Appendix B):

$$\lambda_1 = \frac{3}{8} + \frac{1}{8}\mathrm{sinc}(2\pi c) + \frac{1}{8}\sqrt{(1 - \mathrm{sinc}(2\pi c))^2 + 16\mathrm{sinc}(\pi c)^2}, \qquad \lambda_2 = \frac{1}{4} - \frac{1}{4}\mathrm{sinc}(2\pi c),$$

$$\lambda_3 = \frac{3}{8} + \frac{1}{8}\mathrm{sinc}(2\pi c) - \frac{1}{8}\sqrt{(1 - \mathrm{sinc}(2\pi c))^2 + 16\mathrm{sinc}(\pi c)^2}, \qquad \lambda_4 = 0. \tag{7}$$

For an $n$-qubit system, the largest eigenvalue $\eta_{\max}$ of the kernel in Eq. 6 falls exponentially with $n$ for $c \sim \mathcal{O}_n(1)$ and makes generalization impossible with $P \sim \text{poly}(n)$ amount data (Kübler et al., 2021) (see Appendix. B). To prevent $\eta_{\max}$ decreasing with $n$, we choose a bandwidth that scales with $n$ (i.e., $c = an^{-\alpha}$ with $a \sim \mathcal{O}(1)$ and $\alpha > 0$). In Appendix B we show that $\alpha \geq \frac{1}{2}$ is required for generalization. We consider $\alpha = 1/2$ henceforth. Then, all nonzero eigenvalues of the $n$-qubit kernel Eq. 6 can be expressed as $\lambda_1^{k_1} \lambda_2^{k_2} \lambda_3^{k_3}$ with $k_1 + k_2 + k_3 = n$, where the single-qubit eigenvalues asymptotically (with $n$) look like

$$\lambda_1 \approx 1, \quad \lambda_2 \approx \frac{a^2 \pi^2}{6n}, \quad \lambda_3 \approx \frac{a^4 \pi^4}{180 n^2}. \tag{8}$$

Starting from $k_1 = n$ and $k_2 = k_3 = 0$, the hierarchy of eigenvalues obtained in this way are given in Table 1. We denote each of these eigenvalues as $\eta_{k,z}$ and their corresponding eigenfunctions as $\phi_{k,z}(x)$, where $k$ indexes

Table 1: Hierarchy of eigenvalues based on their scaling. $N(n, k)$ denotes the degeneracy of eigenvalues with scaling $n^{-k}$, and $|n, k\rangle$ denotes the form of the corresponding eigenstates.

| $n^{-k}$ | Degeneracy $N(n, k)$ | Eigenstate $|n, k\rangle$ |
|---|---|---|
| $n^0$ | $1$ | $|\psi_1\rangle^{\otimes n}$ |
| $n^{-1}$ | $\binom{n}{1}$ | $|\psi_1\rangle^{\otimes(n-1)} |\psi_2\rangle$ |
| $n^{-2}$ | $\binom{n}{2} + \binom{n}{1}$ | $|\psi_1\rangle^{\otimes(n-2)} |\psi_2\rangle^{\otimes 2}, |\psi_1\rangle^{\otimes(n-1)} |\psi_3\rangle$ |
| $n^{-3}$ | $\binom{n}{3} + \binom{n-1}{1}\binom{n}{1}$ | $|\psi_1\rangle^{\otimes(n-3)} |\psi_2\rangle^{\otimes 3}, |\psi_1\rangle^{\otimes(n-2)} |\psi_2\rangle |\psi_3\rangle$ |

the overall scaling of the eigenvalue and $z$ indexes each of the individual eigenvalues with scaling $n^{-k}$. The degeneracy $N(n, k) \sim \mathcal{O}(n^k)$ denotes the number of eigenvalues $\eta_{k,z} \sim \mathcal{O}(n^{-k})$ for each $k$. Notice that the quantity $\bar{\eta}_{k,z} = N(n, k)\eta_{k,z} \sim \mathcal{O}_n(1)$ with respect to the input dimension and its value depends on $a$ and $k$. For each $k$, we will further make the approximation $\bar{\eta}_{k,z} \approx \bar{\eta}_{k,z'}$ for all pairs $z, z'$ since the spectrum is almost flat at each scaling $k$ (see Figure 2A).

We obtain the projections of target function $\bar{f}(x)$ on the kernel eigenfunctions as $\bar{a}_{k,z} = \int \bar{f}(x)\phi_{k,z}(x)p(x)dx$. We also define the total target power at each scaling $\bar{a}_k^2 \equiv \sum_{z=1}^{N(n,k)} \bar{a}_{k,z}^2$. With the eigenvalues and the target weights $\bar{a}_k^2$, the generalization error is given by (Canatar et al., 2021) (Appendix C):

$$E_g = \frac{\kappa^2}{1 - \gamma} \sum_k \frac{\bar{a}_k^2}{\left(\kappa + \alpha_k \bar{\eta}_k\right)^2}, \quad \kappa = \lambda + \kappa \sum_k \frac{\bar{\eta}_k}{\kappa + \alpha_k \bar{\eta}_k}, \quad \gamma = \sum_k \frac{\alpha_k \bar{\eta}_k^2}{(\kappa + \alpha_k \bar{\eta}_k)^2}, \tag{9}$$

where $\lambda$ is the KRR regularization parameter, $\kappa$ should be solved self-consistently, and we have defined $\alpha_k = \frac{P}{N(n,k)}$. Taking the large qubit and large data limit ($n \to \infty$ and $P \to \infty$) while keeping $\alpha_l \sim \mathcal{O}_n(1)$ for some mode $l$ (meaning $P \sim \mathcal{O}(n^l)$), the generalization error decouples across different scaling limits and becomes

$$E_g(\alpha_l) = \left( \frac{\kappa^2}{1 - \gamma} \frac{\bar{a}_l^2}{\left(\kappa + \alpha_l \bar{\eta}_l\right)^2} + \frac{\gamma}{1 - \gamma} \sum_{k > l} \bar{a}_k^2 \right) + \sum_{k > l} \bar{a}_k^2. \tag{10}$$

The target modes with $k > l$ remain unlearned since $\alpha_{k>l} = 0$ and the modes $k < l$ have already been learned using the provided data since $\alpha_{k<l} = \infty$. This scaling defines *learning stages* where at each stage a single mode $l$ is being learned, and the remaining modes contribute as constant error. The term in the parentheses goes to zero as $\alpha_l \to \infty$ (see Appendix C). Hence, when the mode $l$ is completely learned, the ratio of the generalization error to its initial value becomes $\frac{E_g(\alpha_l = \infty)}{E_g(0)} = (\sum_{k>l} \bar{a}_k^2)/(\sum_k \bar{a}_k^2)$. Therefore, given a data budget $P \sim \mathcal{O}(n^l)$, the quantum kernel is guaranteed to generalize to target functions whose weights beyond mode $l$ are vanishing. The quantity $C(l) = 1 - \frac{E_g(\alpha_l = \infty)}{E_g(0)}$, called the *cumulative power* (Canatar et al., 2021), describes the amount of power placed in the first $l$ modes and quantifies the task-model alignment.

It was shown in Canatar et al. (2021) that kernels generalize better for target functions with sharply rising $C(l)$. In our case, for generalizability at $P \sim \mathcal{O}(n^l)$ samples, it is a necessary but not sufficient condition for

target functions to have good task-model alignment for which $C(l) \approx 1$. Note that the trace of this kernel is $\int k(\mathbf{x}, \mathbf{x})p(\mathbf{x})d\mathbf{x} = 1$, and therefore the eigenvalues satisfy $\sum_{k,z} \eta_{k,z} = 1$. Flatness of the spectrum, then, implies $\eta_{k,z} \sim \mathcal{O}(3^{-n})$ since there are $3^n$ nonzero modes of the kernel in Eq. 6 (see Appendix B). Equation 9 suggests that even if the target is aligned well with the kernel, it requires $P \sim \mathcal{O}(3^n)$ samples to learn each mode, and so generalization becomes impossible with polynomial sample complexity $P \sim \mathcal{O}(n^l)$.

On the other hand, bandwidth enables sufficient decay in the spectrum of the kernel, and Eq. 9 shows that $P \sim \mathcal{O}(n^l)$ samples yield an excess generalization error $E_g \approx 1 - C(l)$. In Figure 2, we show the results of simulating the kernel of Eq. 6 with $n = 40$ qubits for a target function given by $\bar{f}(\mathbf{x}) = e^{-\|\mathbf{x}\|^2/n^2}$. For $c = 1$, the kernel has a flat spectrum (Figure 2A) with poor alignment with the task (Figure 2B). On the other hand, bandwidth introduces sufficient decay in the eigenspectrum so that polynomial time learning becomes possible. Surprisingly, bandwidth also improves the task-model alignment which, together with spectral bias, implies generalizability with better sample efficiency. In Figure 2C, we perform kernel regression with our toy kernel and confirm that generalization improves with bandwidth up to an optimal value after which it degrades again. This is due to the fact that larger bandwidths cause underfitting of the data (Silverman, 2018) since only a very few eigenmodes become learnable while the target cannot be fully explained by those modes. In Figure 2, we find that the optimal bandwidth parameter is $c^* \approx \frac{2}{n}$ (see Appendix E).

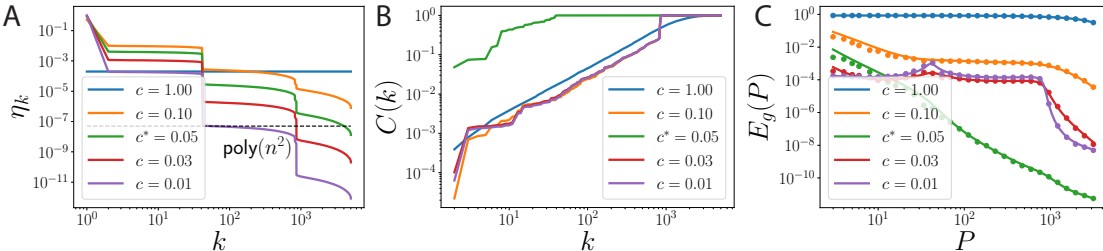

Figure 2: **A** Eigenvalues of the kernel Eq. 6 with respect to data $\mathbf{x} \sim \text{Unif}([-\pi, \pi]^{40})$. Learnability is explained by the preservation of large-eigenvalue eigenspaces; without bandwidth, the model provably cannot learn in poly(n) complexity due to the flat spectrum. **B** The projections of the target $\bar{f}(\mathbf{x}) = e^{-\|\mathbf{x}\|^2/n^2}$ on the eigenvectors of the kernel for each bandwidth. Apart from the flatness of the $c = 1$ kernel, its eigenfunctions align poorly with the target. **C** Generalization error as a function of the number of training samples computed by using theory (solid lines) (Eq. 9) and performing kernel ridge regression empirically (dots). Bandwidth $c = 1$ yields a constant learning curve. While all $c < 1$ kernels provide improvement, an optimal bandwidth parameter $c^* \approx 2/n$ gives the best task-model alignment.

## 4.2 Evidence of performance gains in real datasets

To evaluate our theory in a practical setting, we consider two previously proposed quantum kernels that have been conjectured to be hard to simulate classically: a kernel with a feature map inspired by instantaneous quantum polynomial-time (IQP) circuit (Shepherd & Bremner, 2009; Havlíček et al., 2019; Huang et al., 2021) and a kernel with Hamiltonian evolution (EVO) feature map (Huang et al., 2021; Shaydulin & Wild, 2021) (see Appendix E for details). Unlike the kernel considered in the preceding section, the spectrum cannot be derived analytically for these kernels.

We evaluate these kernels on binary classification versions of FMNIST (Xiao et al., 2017), KMNIST (Clanuwat et al., 2018), and PLAsTiCC (The PLAsTiCC team et al., 2018) datasets with the input data downsized to $n = 22$ dimensions, which were previously used to evaluate quantum kernel performance in Shaydulin & Wild (2021); Huang et al. (2021); Peters et al. (2021). We use the kernel values reported in Shaydulin & Wild (2021), which were evaluated with high precision using an idealized (noiseless) simulator. In practice, an additive error is introduced when evaluating the kernel values on a fault-tolerant quantum computer. Bandwidth-equipped kernels are robust against this error; see the discussion in Shaydulin & Wild (2021). We perform SVM for binary classification using these kernels with varying bandwidths. In Table 2, we report the test accuracies with bandwidth parameter $c = 1$ for the IQP and EVO kernels. We also report the test

performance with bandwidth parameters $c^* < c$ optimized by hyperparameter tuning for each kernel using cross validation (see Appendix E). For both kernels, bandwidth significantly improves the test performance.

| | IQP | | EVO | | Random Guess |
|---|---|---|---|---|---|
| | $c^*$ | $c = 1$ | $c^*$ | $c = 1$ | |
| FMNIST | 0.926 | 0.542 | 0.916 | 0.643 | 0.5 |
| KMNIST | 0.915 | 0.629 | 0.914 | 0.600 | 0.5 |
| PLAsTiCC | 0.789 | 0.5 | 0.788 | 0.613 | 0.5 |

Table 2: Bandwidth tuning recovers significant performance gains over out-of-the-box quantum models, opening up the possibility of better workflows for general quantum machine learning on many qubits via hyperparameter tuning.

To test our intuition, we present in Fig. 3A the shape of the IQP kernel where the kernel clearly sharpens for larger values of bandwidth parameter implying flat spectrum. In Fig. 3B, we further confirm that the spectrum decay improves with bandwidth when the IQP kernel is evaluated on the FMNIST dataset. We also find that the task-model alignment improves greatly with the bandwidth (Fig. 3C), thus implying, together with the previous point, possible generalization.

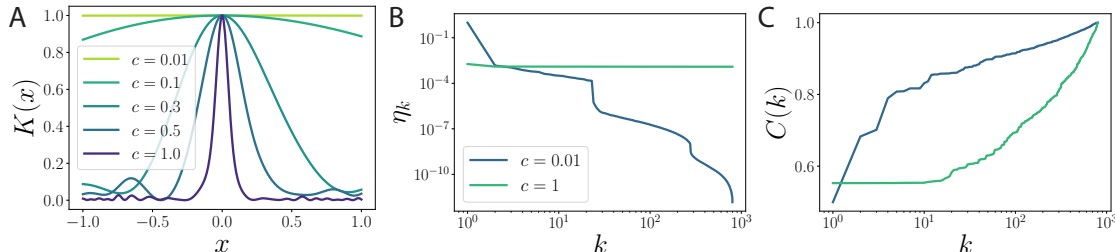

Figure 3: **A** IQP kernel function example: Intuitive behavior of bandwidth persists even when quantum kernel is not strictly shift-invariant. **B** IQP with the addition of bandwidth is capable of recovering significant target alignment with FMNIST dataset. **C** While the spectrum for $c = 1$ IQP is flat, it also has poor alignment with the target (see Appendix E).

## 5 Related Work

Huang et al. (2021) introduce the idea that the exponential dimensionality of quantum feature spaces precludes generalization of quantum kernel methods, and they provide an upper bound on generalization error that includes the dimension of the space spanned by the training set. They connect their results to the spectrum of the kernel by introducing a measure of "approximate dimension" of the span of the training set given by $\sum_{k=1}^{n} \left( \frac{1}{n-k} \sum_{l=k}^{m} t_l \right)$, where $\{t_l\}_{l=1}^{n}$ are the eigenvalues of the $n \times n$ kernel matrix. This number is effectively a measure of the flatness of the spectrum of the kernel and is used in numerical experiments in Huang et al. (2021). An alternative analysis is given by Banchi et al. (2021), who derive bounds on generalization error using quantum information techniques and show that near-identity kernels resulting from large dimensionality of the feature space preclude generalization. We use the construction from Huang et al. (2021, Appendix I) as our motivating example in Sec. 3.

Kübler et al. (2021) introduce techniques for deriving the spectrum of quantum kernels and obtain the spectrum of the kernel Eq. 5. They then use these techniques to show that the purity of the mean embedding provides an upper bound on the largest eigenvalue of the quantum kernel integral operator and consequently that quantum kernel methods with feature maps that utilize the full quantum Hilbert space require an exponential number of qubits of data to learn. However, Kübler et al. (2021) did not consider bandwidth as a method for controlling the inductive bias of quantum kernels. Our contribution is using the techniques of

Kübler et al. (2021) to derive the spectrum of the bandwidth-equipped kernel Eq. 6 and explicit formulas for the expected generalization error.

Inspired by the bandwidth hyperparameter of classical radial basis function kernels, Shaydulin & Wild (2021) introduce the concept of quantum kernel bandwidth and provide numerical evidence that bandwidth tuning (equivalent to rescaling of the input data) improves the generalization of SVM with quantum kernels. An analogous mechanism has been observed to improve trainability of quantum neural networks (Zhang et al., 2022). However, previous results make no connection between the bandwidth and the spectrum of the kernel and provide no analytical results for generalization. We reinterpret the data from Shaydulin & Wild (2021) in Sec. 4.2 and show how the bandwidth controls the spectrum of the kernel and enables generalization.

## 6 Discussion

In this work, we studied the kernels induced by quantum feature embeddings of data and their generalization potential. Recent work suggests that machine learning models built in this way suffer from the curse of dimensionality, requiring exponentially large training sets to learn. Note that embedding $n$-dimensional data requires at least $n$ qubits (where $n \sim 10^3$ for standard datasets) which span a $2^n$ dimensional feature space. While quantum models may possess potentially powerful and classically inaccessible representations for certain tasks, utilization of those necessarily requires a control over the large space they span in order to generalize.

Here we showed that the bandwidth hyperparameter enables generalization of quantum kernel methods when the numbers of qubits is large, and provided explicit formulas for the resulting expected generalization error on a toy model. Our results open up promise for quantum machine learning beyond intermediate numbers of qubits. A central lesson provided by our work is that thoughtfully chosen hyperparameters can significantly improve the performance of quantum machine learning methods. Identifying such hyperparameters that control the inductive bias of quantum models is essential to realizing the full potential of quantum machine learning methods.

Unlike prior works, we focus not just on the spectrum of the quantum kernel but on the alignment between the kernel and the real-world datasets. Our empirical results imply that scaling the bandwidth not only makes the spectrum less flat, but also improves the alignment with real-world target functions. These observations make us optimistic about the potential of quantum kernel methods to solve classically challenging problems.

An important limitation of our results is that while bandwidth scaling is guaranteed to improve the spectrum, it does not necessarily lead to good alignment. For example, in the limit of $c \to 0$ the spectrum has only one nonzero eigenvalue, although learning is still not possible (see Appendix D). This suggests that optimizing bandwidth as a hyperparameter during training can balance triviality of the feature map ($c \to 0$) with greater utilization of the quantum state space. At the same time, if the feature map is chosen poorly, varying bandwidth would not lead to good generalization.

If the kernel values are evaluated on a noisy quantum computer, the quantum hardware noise would affect the spectrum of the kernel. Hardware noise reduces the purity of embedding, leading to a trivial lower bound on generalization error from Kübler et al. (2021, Theorem 1). Heyraud et al. (2022) and Wang et al. (2021) give a more detailed analysis. While outside the scope of this work, the impact of noise on generalization is nonetheless an important consideration for near-term prospects of quantum kernel methods.

**Acknowledgments**

Funding in direct support of this work: U.S. Department of Energy (DOE), Office of Science, Office of Advanced Scientific Computing Research AIDE-QC and FAR-QC projects and by the Argonne LDRD program under contract numbers DE-AC02-05CH11231 and DE-AC02-06CH11357; computing resources provided on Bebop, a high-performance computing cluster operated by the Laboratory Computing Resource Center at Argonne National Laboratory. AC and CP were supported by an award from the Harvard Data Science Initiative Competitive Research Fund.

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

**Disclaimer**

## A Review of Classical and Quantum Data Operators

Here we discuss covariance operators and integral kernel operators for two types of data: classical real-valued vectors and finite-dimensional complex Hermitian operators. The goal is to relate the spectra of these operators in order to better understand how feature maps and distributions of input data combine to affect the learnability of a distribution.

Given a symmetric positive semidefinite kernel function $k : \mathcal{X} \times \mathcal{X} \to \mathbb{R}$, the integral kernel operator $T_k : L_2(\mathcal{X}) \to L_2(\mathcal{X})$ is defined according to its action on $f \in L_2(\mathcal{X})$ as

$$(T_k f)(\mathbf{x}) = \int_{\mathcal{X}} k(\mathbf{x}, \mathbf{x}') f(\mathbf{x}') \mu(d\mathbf{x}'). \tag{A.1}$$

A quantum kernel is defined as the inner product of two quantum states,

$$k(\mathbf{x}, \mathbf{x}') = \langle \rho(\mathbf{x}), \rho(\mathbf{x}') \rangle_{\mathcal{H}} = \mathrm{Tr}\{\rho(\mathbf{x})\rho(\mathbf{x}')\} = \mathrm{Vec}\left(\rho(\mathbf{x})\right)^{\dagger} \mathrm{Vec}\left(\rho(\mathbf{x}')\right),$$

where we have introduced the vectorization map (e.g., Watrous (2018)) $\mathrm{Vec} : L(\mathbb{C}^d) \to \mathbb{C}^{d^2}$, which stacks rows of a $d \times d$ matrix into a $d^2 \times 1$ column vector and where $L(\mathbb{C}^d)$ denotes the space of linear operators of the form $\mathbb{C}^d \to \mathbb{C}^d$. Throughout this section we will consider $d$-dimensional quantum states and operations (where $d = 2^n$ in the case of $n$ qubits). We will frequently use the identity

$$\mathrm{Vec}\,(A)^{\dagger} \mathrm{Vec}\,(B) = \langle A, B \rangle = \mathrm{Tr}\left\{A^{\dagger}B\right\}, \qquad \forall A, B \in L(\mathbb{C}^d).$$

Observing that the RKHS of a kernel $k$ is defined (see, e.g., Schuld (2021)) by functions of the form

$$f(x) = \langle \rho(\mathbf{x}), H \rangle,$$

we can rewrite Eq. A.1 as

$$\begin{aligned}
(T_k f)(x) &= \mathrm{Vec}\left(\rho(\mathbf{x})\right)^{\dagger} \int_{\mathcal{X}} \mathrm{Vec}\left(\rho(\mathbf{x}')\right) \mathrm{Vec}\left(\rho(\mathbf{x}')\right)^{\dagger} \mathrm{Vec}\,(H)\,\mu(d\mathbf{x}') \\
&= \langle \mathrm{Vec}\left(\rho(\mathbf{x})\right), \Sigma\,\mathrm{Vec}\,(H) \rangle,
\end{aligned} \tag{A.2}$$

where we have defined the *covariance operator*,

$$\Sigma = \int_{\mathcal{X}} \mathrm{Vec}\left(\rho(\mathbf{x}')\right) \mathrm{Vec}\left(\rho(\mathbf{x}')\right)^{\dagger} \mu(d\mathbf{x}') = \int_{\mathcal{X}} \rho(\mathbf{x}') \otimes \rho(\mathbf{x}')^{T} \mu(d\mathbf{x}'), \tag{A.3}$$

and the last equality in Eq. A.2 holds whenever the quantum embedding is a pure state (i.e., $\rho(\mathbf{x}) = |\psi(\mathbf{x})\rangle\langle\psi(\mathbf{x})|$ for some $|\psi(\mathbf{x})\rangle$). We can intuitively understand this operator by analogy to linear feature maps: If we consider $n$-dimensional real-valued input vectors $\mathbf{x} \in \mathcal{X} \subset \mathbb{R}^n$ and assume $\mathbf{x}$ are centered such that $\mathbf{E}_{\mathcal{X}}[\mathbf{x}] = 0$, then the covariance operator is defined as $\Sigma = \mathbf{E}_{\mathcal{X}}[\mathbf{x}\mathbf{x}^T]$ such that $(\Sigma)_{ij} = \mathbf{E}[x_i x_j]$. Equation A.3 therefore describes a kind of quantum covariance operator, although this is an imperfect analogy since a centered quantum feature map ($\mathbf{E}_{\mathcal{X}}[\rho(\mathbf{x})] = 0$) would violate the requirement $\mathrm{Tr}\{\rho(\mathbf{x})\} = 1$.

We are interested in the eigenvalue equations

$$\Sigma\,\mathrm{Vec}\,(H) = \eta\,\mathrm{Vec}\,(H)$$
$$(T_k \phi)(\mathbf{x}) = \eta \phi(\mathbf{x}).$$

Given the operators defined in Eq. A.1 and Eq. A.3, one may verify (see, e.g., Shawe-Taylor et al. (2005)) that the following statements hold under the identification $\psi(\mathbf{x}) \to \mathrm{Vec}\left(\rho(\mathbf{x})\right)$ described above.

(S1) For every eigenfunction $\phi$ satisfying $(T_k \phi)(\mathbf{x}) = \eta \phi(\mathbf{x})$, there is a corresponding eigenvector $\mathrm{Vec}\,(H)$ of $\Sigma$ given by

$$\mathrm{Vec}\,(H) = \int_{\mathcal{X}} \phi(\mathbf{x}) \mathrm{Vec}\left(\rho(\mathbf{x})\right) \mu(d\mathbf{x})$$

such that $\Sigma\,\mathrm{Vec}\,(H) = \eta\,\mathrm{Vec}\,(H)$. Furthermore, from the bijectivity of the Vec operation and the fact that Hermiticity is preserved under convex combination, it follows that $H$ is Hermitian.

(S2) For every eigenvector $\mathrm{Vec}\,(H)$ of $\Sigma$ satisfying $\Sigma\,\mathrm{Vec}\,(H) = \eta\,\mathrm{Vec}\,(H)$, there is a corresponding eigenfunction of $T_k$ given by

$$\phi(\mathbf{x}) = \mathrm{Tr}\big\{\rho(\mathbf{x})H\big\} = \mathrm{Vec}\,\big(\rho(\mathbf{x})\big)^\dagger \,\mathrm{Vec}\,(H)$$

such that $(T_k\phi)(\mathbf{x}) = \eta\phi(\mathbf{x})$.

(S3) Assuming that eigenfunctions $\phi_k$ of $T_k$ indexed by eigenvalue $\eta_k$ are orthonormal,

$$\langle\phi_k, \phi_l\rangle_{L_2(\mathcal{X})} = \int_{\mathcal{X}} \phi_k(\mathbf{x})^* \phi_l(\mathbf{x})\mu(d\mathbf{x}) = \delta_{kl},$$

the eigenvectors $\mathrm{Vec}\,(H_k)$ of $\Sigma$ indexed according to eigenvalue $\eta_k$ satisfy

$$\mathrm{Vec}\,(H_k)^\dagger \,\mathrm{Vec}\,(H_l) = \mathrm{Tr}\Big\{H_k^\dagger H_l\Big\} = \eta_k^{-1}\delta_{kl}$$

whenever $\eta_k > 0$ (when $\eta_k = 0$, we may safely ignore $\mathrm{Vec}\,(H_k)$ in the null space of $\Sigma$).

Letting $\mathrm{spec}\,(A)$ denote the sequence of eigenvalues of an operator $A$ sorted in nonincreasing order, it then follows from (S1) and (S2) that

$$\mathrm{spec}\,(T_k) = \mathrm{spec}\,(\Sigma)\,.$$

The *mean embedding* is defined as the average state with respect to a distribution over $\mathcal{X}$:

$$\rho_\mu = \int_{\mathcal{X}} \rho(\mathbf{x})\mu(d\mathbf{x}).$$

As shown in Kübler et al. (2021, Lemma 1), the inequality $\eta_{\max}(\Sigma) \leq \sqrt{\mathrm{Tr}\{\rho_\mu\}^2}$ provides a bound for the largest eigenvalue of $\Sigma$. Let $\mathbf{n}$ be the Bloch vector for $\rho_\mu$ defined on $n = 1$ qubits, which can be parameterized as

$$\mathbf{n} = (\sin\theta\cos\phi, \sin\theta\sin\phi, \cos\phi) \tag{A.4}$$

Then $\mathrm{Tr}\Big\{\rho_\mu^2\Big\} = (1 + \|\mathbf{n}\|^2)/2$. This provides a geometric argument for the use of bandwidth in a single-qubit system (or product thereof). For instance, consider each shaded region in Fig. 1A representing the subspace spanned by quantum states associated with the feature map giving rise to $k$. Then we have that the maximum eigenvalue of $\Sigma$ is suppressed like $\eta_{\max}(\Sigma) \leq (1 + \|\mathbf{n}\|^2)^{1/2}/\sqrt{2}$ as the centroid $\mathbf{n}$ of each region (representing $\rho_\mu$) approaches the center of the Bloch sphere. By limiting bandwidth (and therefore restricting the shaded region of Fig. 1B to a polar cap of the Bloch sphere), the upper bound on $\eta_{\max}(\Sigma)$ is lifted, and the possibility of learning on data is restored.

## A.1 Spectrum of a random quantum embedding

A foundational observation for this work is that using a kernel associated with a feature map that completely utilizes a high-dimensional feature space leads to poor learning guarantees due to the flatness of the corresponding spectrum. Here, we present an extreme (and somewhat contrived) example where a quantum feature map associated with random state vectors leads to a flat kernel spectrum. We assume the existence of a data distribution and feature map $\mathbf{x} \to U(\mathbf{x})|0\rangle\langle 0|U^\dagger(\mathbf{x})$ for which unitaries $U(\mathbf{x})$ are sampled uniformly over the space of $n$-qubit unitaries $U(2^n)$. Such a distribution may be achieved by sampling uniformly with respect to the Haar measure $\mu(dU)$ (e.g., Collins & Nechita (2016)). As described in Rosasco et al. (2010, Proposition 9), the spectrum of $\mathbf{K}$ concentrates around the spectrum of $\Sigma$. We therefore explicitly compute the spectrum of $\Sigma$ corresponding to random quantum features. Using standard results for integration with

respect to the Haar measure (Puchała & Miszczak, 2017), we compute the average elements of $\Sigma$ as

$$
\begin{aligned}
\langle ij|\Sigma|k\ell\rangle &= \int_{U(2^n)} \langle i|\rho(\mathbf{x})|k\rangle\langle j|\rho(\mathbf{x})|\ell\rangle\mu(dU) \\
&= \int_{U(2^n)} U_{i0}U_{k0}^* U_{j0}U_{\ell 0}^* \mu(dU) \\
&= \frac{2^n - 1}{2^n(2^{2n} - 1)}\left(\delta_{ik}\delta_{j\ell} + \delta_{i\ell}\delta_{jk}\right).
\end{aligned}
\tag{A.5}
$$

The terms associated with $\delta_{ik}\delta_{j\ell} = 1$ correspond to the identity operator $\mathrm{I}_{4^n}$, while the terms $\delta_{i\ell}\delta_{jk}$ contribute to either $2^n$-many diagonal components $\Sigma_{ii,ii}$ for $i = 1, \ldots, 2^n$ or $2^n(2^n - 1)$-many off-diagonal components $\Sigma_{ij,ji}$ whenever $j \neq i$:

$$
\Sigma \to \mathrm{I}_{4^n} + \sum_{i=1}^{2^n} |ii\rangle\langle ii| + \sum_{i=1}^{2^n} \sum_{j \neq i} |ij\rangle\langle ji|.
\tag{A.6}
$$

Equation A.6 is therefore a direct sum of subspaces proportional to $2\mathrm{I}_2$ and subspaces proportional to $\mathrm{I}_2 + \mathrm{X}_2$, the latter of which may be easily diagonalized. Combining with Eq. A.5 yields the spectrum

$$
\mathrm{spec}\left(\Sigma\right) = \begin{cases} \frac{2}{2^n(2^n+1)} & \text{with multiplicity } 2^n + \frac{2^n(2^n-1)}{2} \\ 0 & \text{with multiplicity } \frac{2^n(2^n-1)}{2}. \end{cases}
$$

Coincidentally, the mean embedding associated with this feature map is proportional to the identity; that is, $\rho_\mu = \mathbf{E}_{\mathcal{X}}[\rho(\mathbf{x})] = \mathrm{I}/2^n$.

## B  Spectrum of the Bandwidth-Equipped Kernel

Here we derive the spectrum of the integral operator for kernel discussed in the main text. We follow the technique of diagonalizing $\Sigma$ described in Appendix A and used in Kübler et al. (2021, Appendix C). We first derive the spectrum for the single-qubit (one-dimensional data) case. Then we leverage the observation that the kernel factorizes over the qubits to obtain the full spectrum for the general $n$-dimensional case.

We begin by considering the input $x \in \mathrm{Uniform}[-\pi, \pi]$. Recall that the feature map $\rho(x)$ is generated by the unitary

$$
U(x) = \cos\left(\frac{cx}{2}\right)\mathrm{I} + i\sin\left(\frac{cx}{2}\right)\mathrm{X}.
$$

Then, the vectorization of the image of a data point $x$ in feature space $\rho(x) = U(x)|0\rangle\langle 0|U(x)^\dagger$ is

$$
\mathrm{Vec}\left(\rho(x)\right) = \begin{pmatrix} \cos^2\left(\frac{cx}{2}\right) \\ -i\cos\left(\frac{cx}{2}\right)\sin\left(\frac{cx}{2}\right) \\ i\cos\left(\frac{cx}{2}\right)\sin\left(\frac{cx}{2}\right) \\ \sin^2\left(\frac{cx}{2}\right) \end{pmatrix},
$$

and the corresponding kernel is

$$
k(x, x') = \mathrm{Vec}\left(\rho(x)\right)^\dagger \mathrm{Vec}\left(\rho(x')\right) = \cos^2\left(c\frac{(x - x')}{2}\right).
\tag{A.7}
$$

Our goal is to compute the eigenvalues of the integral operator $T_k$ defined as

$$
(T_k\phi_k)(x) = \int \mu(\mathrm{d}x')k(x, x')\phi_k(x') = \lambda_k\phi_k(x).
\tag{A.8}
$$

Here we refer to single-qubit eigenvalues as $\lambda_k$ and many-qubit eigenvalues as $\eta_k$. As described in Appendix A, this is equivalent to computing the eigenvalues of the covariance operator $\Sigma$ given by

$$
\Sigma = \int_{\mathcal{X}} \operatorname{Vec}\left(\rho(y)\right) \operatorname{Vec}\left(\rho(y)\right)^{\dagger} \mu(\mathrm{d}y)
$$

$$
= \frac{1}{2\pi} \int_{-\pi}^{\pi}
\begin{pmatrix}
\cos^4\left(\frac{cy}{2}\right) & i\cos^3\left(\frac{cy}{2}\right)\sin\left(\frac{cy}{2}\right) & -i\cos^3\left(\frac{cy}{2}\right)\sin\left(\frac{cy}{2}\right) & \cos^2\left(\frac{cy}{2}\right)\sin^2\left(\frac{cy}{2}\right) \\
-i\cos^3\left(\frac{cy}{2}\right)\sin\left(\frac{cy}{2}\right) & \cos^2\left(\frac{cy}{2}\right)\sin^2\left(\frac{cy}{2}\right) & -\cos^2\left(\frac{cy}{2}\right)\sin^2\left(\frac{cy}{2}\right) & -i\cos\left(\frac{cy}{2}\right)\sin^3\left(\frac{cy}{2}\right) \\
i\cos^3\left(\frac{cy}{2}\right)\sin\left(\frac{cy}{2}\right) & -\cos^2\left(\frac{cy}{2}\right)\sin^2\left(\frac{cy}{2}\right) & \cos^2\left(\frac{cy}{2}\right)\sin^2\left(\frac{cy}{2}\right) & i\cos\left(\frac{cy}{2}\right)\sin^3\left(\frac{cy}{2}\right) \\
\cos^2\left(\frac{cy}{2}\right)\sin^2\left(\frac{cy}{2}\right) & i\cos\left(\frac{cy}{2}\right)\sin^3\left(\frac{cy}{2}\right) & -i\cos\left(\frac{cy}{2}\right)\sin^3\left(\frac{cy}{2}\right) & \sin^4\left(\frac{cy}{2}\right)
\end{pmatrix} \mathrm{d}y
$$

$$
=
\begin{pmatrix}
a_1 & 0 & 0 & a_2 \\
0 & a_2 & -a_2 & 0 \\
0 & -a_2 & a_2 & 0 \\
a_2 & 0 & 0 & a_3
\end{pmatrix},
$$

where

$$
a_1 = \frac{3}{8} + \frac{1}{2}\operatorname{sinc}(c\pi) + \frac{1}{8}\operatorname{sinc}(2c\pi)
$$

$$
a_2 = \frac{1}{8} - \frac{1}{8}\operatorname{sinc}(2c\pi)
$$

$$
a_3 = \frac{3}{8} - \frac{1}{2}\operatorname{sinc}(c\pi) + \frac{1}{8}\operatorname{sinc}(2c\pi).
$$

We can easily obtain the eigenvalues of this matrix, which are

$$
\lambda_1 = \frac{3}{8} + \frac{1}{8}\operatorname{sinc}(2\pi c) + \frac{1}{8}\sqrt{(1 - \operatorname{sinc}(2\pi c))^2 + 16\operatorname{sinc}(\pi c)^2}
$$

$$
\lambda_2 = \frac{1}{4} - \frac{1}{4}\operatorname{sinc}(2\pi c)
$$

$$
\lambda_3 = \frac{3}{8} + \frac{1}{8}\operatorname{sinc}(2\pi c) - \frac{1}{8}\sqrt{(1 - \operatorname{sinc}(2\pi c))^2 + 16\operatorname{sinc}(\pi c)^2}
$$

$$
\lambda_4 = 0, \tag{A.9}
$$

where $\lambda_1 > \lambda_2 \geq \lambda_3 > \lambda_4$ for all $c \in [0,1]$. With $c = 1$, the eigenvalues become $\frac{1}{2}, \frac{1}{4}, \frac{1}{4}, 0$, respectively. Now we can examine the impact of bandwidth on the eigenvalues (see Eq. A.9) of the integral operator. We first observe that for $c \to 0$, all eigenvalues except the top eigenvalue become zero; that is, $\lambda_1 \to 1$ and $\lambda_2, \lambda_3, \lambda_4 \to 0$. This confirms our intuition about the impact of bandwidth on the spectrum of the integral operator. This also implies that for small $c$ the approximate dimension of the space spanned by the training data will be 1, which is consistent with the observation that in this limit the feature maps become constant.

For an $n$-qubit system and an input data distribution that factorizes over the dimensions, the kernel simply becomes the direct product of the $n$ copies of the single-qubit system. Since the $n$ qubits are completely decoupled, the resulting kernel in Eq. 6 has eigenvalues of the form

$$
\eta_{n_1 n_2 n_3 n_4} = \lambda_1^{n_1} \lambda_2^{n_2} \lambda_3^{n_3} \lambda_4^{n_4}, \quad n_1 + n_2 + n_3 + n_4 = n, \quad n_1, n_2, n_3, n_4 \in \mathbb{Z}^+ \cup \{0\},
$$

where the nonzero eigenvalues are obtained by setting $n_4 = 0$ since $\lambda_4 = 0$. Here, we note that the number of zero eigenvalues grows exponentially with number of qubits as $4^n - 3^n$. However, its ratio to the total number of eigenvalues vanishes since

$$
\frac{\#\{\eta_{n_1 n_2 n_3 n_4} = 0\}}{\#\{\eta_{n_1 n_2 n_3 n_4}\}} = \frac{4^n - 3^n}{4^n} \xrightarrow{n \to \infty} 0;
$$

therefore the bulk of the spectrum remains nonzero.

For $c \sim \mathcal{O}_n(1)$, the spectrum remains flat as $n \to \infty$. This can be easily seen with the case $c = 1$ where the eigenvalues are given by

$$
\eta_k = 2^{-n} 2^{-k},
$$

where $k = n_2 + n_3$ and takes values in $\{0, \dots, n\}$. Each eigenvalue $\eta_k$ is degenerate with $N(n, k) = 2^k \binom{n}{k}$, and of course $\sum_{k=0}^{n} N(n, k) = 3^n$ gives the number of nonzero eigenvalues. To obtain a nonflat spectrum, we need eigenvalues to scale with the number of qubits $n$. In the next section, we do so by imposing scaling conditions on the eigenvalues.

For completeness, we also provide the *unnormalized* eigenfunctions of the kernel using the eigenvectors of $\Sigma$. Inverting the Vec operation, we get the matrices

$$H_1 = \begin{pmatrix} \frac{4\mathrm{sinc}(\pi c) + \sqrt{(1 - \mathrm{sinc}(2\pi c))^2 + 16\mathrm{sinc}(\pi c)^2}}{1 - \mathrm{sinc}(2\pi c)} & 0 \\ 0 & 1 \end{pmatrix}, \quad H_2 = \begin{pmatrix} 0 & -1 \\ 1 & 0 \end{pmatrix}$$

$$H_3 = \begin{pmatrix} \frac{4\mathrm{sinc}(\pi c) - \sqrt{(1 - \mathrm{sinc}(2\pi c))^2 + 16\mathrm{sinc}(\pi c)^2}}{1 - \mathrm{sinc}(2\pi c)} & 0 \\ 0 & 1 \end{pmatrix}, \quad H_4 = \begin{pmatrix} 0 & 1 \\ 1 & 0 \end{pmatrix}.$$

The corresponding eigenfunctions are given by $\phi_1(x) = \mathrm{Tr}\big(\rho(x) H_i\big)$ and become

$$\phi_1(x) = \sin^2\left(\frac{cx}{2}\right) + \cos^2\left(\frac{cx}{2}\right) \frac{4\mathrm{sinc}(\pi c) + \sqrt{(1 - \mathrm{sinc}(2\pi c))^2 + 16\mathrm{sinc}(\pi c)^2}}{1 - \mathrm{sinc}(2\pi c)}$$

$$\phi_2(x) = i\sin(cx)$$

$$\phi_3(x) = \sin^2\left(\frac{cx}{2}\right) + \cos^2\left(\frac{cx}{2}\right) \frac{4\mathrm{sinc}(\pi c) - \sqrt{(1 - \mathrm{sinc}(2\pi c))^2 + 16\mathrm{sinc}(\pi c)^2}}{1 - \mathrm{sinc}(2\pi c)}$$

$$\phi_4(x) = 0.$$

Setting $c = 1$, we obtain the eigenfunctions given in Kübler et al. (2021): $\phi_1(x) = 1$, $\phi_2(x) = i\sin(x)$, $\phi_3(x) = -\cos(x)$, and $\phi_4 = 0$. Note that unlike the $c = 1$ case, the eigenfunction $\phi_1(x)$ in general is not constant.

## B.1 Scaling restrictions to the bandwidth

The argument given by Kübler et al. (2021, Theorem 1) is that when the largest eigenvalue of the kernel is sufficiently small compared with the sample size, the generalization error is lower bounded by the $L_2$ norm of the target function with probability at least $1 - \epsilon$ as

$$E_g \geq (1 - \epsilon)\left\|\bar{f}\right\|^2 = (1 - \epsilon)\sum_k \bar{a}_k^2,$$

which matches our result from Sec. 4. The result in Kübler et al. (2021) depends on the exponentially small largest eigenvalue of the kernel compared with the amount of training samples. In fact, from Eq. 9 it easily follows that for a polynomial number of training samples $P \sim n^l$ and exponentially suppressed largest eigenvalue $\eta_{\max} \sim 2^{-n}$, the learning is impossible as $n \to \infty$. Kübler et al. (2021, Lemma 1) proves that the largest eigenvalue of the kernel is upper bounded by the so-called purity:

$$\eta_{\max} \leq \sqrt{M_\mu},$$

where purity is given by $M_\mu = \int \mu(d\mathbf{x})\mu(d\mathbf{x}')k(\mathbf{x}, \mathbf{x}')$. We also demonstrate their proof here for the reader's convenience. Consider a normalized kernel satisfying $k(\mathbf{x}, \mathbf{x}) = 1$ for all $\mathbf{x}$. The normalized eigenfunction $\phi_{\max}(\mathbf{x})$ corresponding to the largest eigenvalue $\eta_{\max}$ is $L_2$ bounded by the constant function $\eta_{\max}^{-1/2}\mathbf{1}(\mathbf{x})$ since

$$1 = k(\mathbf{x}, \mathbf{x}) > \eta_{\max}\phi_{\max}(\mathbf{x})^2.$$

Here $\mathbf{1}(\mathbf{x}) = 1$ is the constant function. This immediately implies that

$$\eta_{\max} = \int \mu(d\mathbf{x})\mu(d\mathbf{x}')k(\mathbf{x}, \mathbf{x}')\phi_{\max}(\mathbf{x})\phi_{\max}(\mathbf{x}')$$

$$\leq \eta_{\max}^{-1} \int \mu(d\mathbf{x})\mu(d\mathbf{x}')k(\mathbf{x}, \mathbf{x}')\mathbf{1}(\mathbf{x})\mathbf{1}(\mathbf{x}') = \eta_{\max}^{-1}M_\mu.$$

Given this bound, we demand that the bandwidth should scale such that the purity stays constant with respect to the number of qubits $n$. For our example kernel, this condition translates into

$$M_\mu = \frac{1}{2^n} \left(1 + \text{sinc}(\pi c(n))\right)^n.$$

Inverting this equation is not possible. However, its numerical solution yields a scaling for bandwidth as

$$c(n) = \frac{a}{\sqrt{n}}, \tag{A.10}$$

where $a \sim \mathcal{O}_n(1)$ depends on the fixed purity $M_\mu$. Note that this is a lower bound for the bandwidth to keep purity from inversely scaling with $n$. In principle, we could allow purity to increase with $n$ depending on the target function. For example, $c(n) = an^{-\infty} \to 0$ yields perfect purity since there is only a single mode. Hence, we conclude that the bandwidth should at least scale as

$$c = an^{-\alpha}, \quad \alpha \geq \frac{1}{2}.$$

We remark, however, that the spectrum will collapse to a single mode for large exponents $\alpha$ and generalization will not be possible except for very specific target functions.

## B.2 Scaling of eigenvalues with bandwidth

Using the bandwidth scaling derived in Eq. A.10, we now study the scaling of eigenvalues at the large qubit limit $n \to \infty$. Asymptotically, the $k$th power of these eigenvalues looks like

$$\lambda_1^k \approx 1 - \frac{a^2\pi^2}{6n}k + \mathcal{O}\left(\frac{1}{n^2}\right)$$

$$\lambda_2^k \approx \left(\frac{a^2\pi^2}{6n}\right)^k \left(1 - \frac{a^2\pi^2}{5n}k + \mathcal{O}\left(\frac{1}{n^2}\right)\right)$$

$$\lambda_3^k \approx \left(\frac{a^4\pi^4}{180n^2}\right)^k \left(1 + \frac{a^2\pi^2}{42n}k + \mathcal{O}\left(\frac{1}{n^2}\right)\right).$$

Notice that with this scaling of the bandwidth parameter, the largest eigenvalue remains constant asymptotically. We also remark that eigenvalues composed of a large number ($k \approx n$) of $\lambda_2$ and $\lambda_3$ scale as $e^{-n\log n}$, and we consider them decoupled since none of these modes can be learned with a polynomial amount of data. Hence, we conclude that the spectral bias induced by the bandwidth restricts the space of learnable functions to lie in the space spanned by eigenfunctions of the form

$$|\psi_1\rangle^{\otimes(n-n_2-n_3)} |\psi_2\rangle^{\otimes n_2} |\psi_3\rangle^{\otimes n_3},$$

such that $n_2 + n_2 \ll n$. It can be shown that the hierarchy of eigenvalues obtained in this way scale polynomially with $n$, as discussed in Sec. 4. In Table B.2, we present the first few eigenvalue scalings where $N(n,k)$ denotes the number of eigenvalues with scaling $\eta_{k,z} \sim \mathcal{O}(n^{-k})$ and $|\Psi\rangle$ denotes the corresponding states. We denote each eigenvalue $\eta_{k,z}$ with two indices: $k$ corresponds to the scaling of the eigenvalue as $n^{-k}$, and $z = 1, \ldots, N(n,k)$ indexes the $N(n,k)$ eigenvalues with the same scaling.

## B.3 Bandwidth and projected (biased) kernels

An alternative way to control the inductive bias of the quantum model is to define the kernel in terms of the reduced density matrix (e.g., single-qubit): $\tilde{\rho}(x) = \text{Tr}_{[1\ldots n-1]}(\rho(x))$, where $\text{Tr}_{[1\ldots n-1]}(\cdot)$ denotes the partial trace over qubits $1, \ldots, n-1$ of a $n$-qubit system. For a detailed discussion of such kernels, the interested reader is referred to Kübler et al. (2021); Huang et al. (2021). Here, we briefly comment on the similarities and differences between the impacts of projection and bandwidth tuning on the spectrum of the kernel.

Table 3: Degeneracies $N(n,k)$ of quantum states for the first few scalings.

| Degenerate States and Spectrum Scaling | | |
|---|---|---|
| Scaling $n^{-k}$ | Degeneracy $N(n,k)$ | States $\lvert\Psi\rangle$ |
| $n^0$ | $1$ | $\lvert\psi_1\rangle^{\otimes n}$ |
| $n^{-1}$ | $\binom{n}{1}$ | $\lvert\psi_1\rangle^{\otimes(n-1)}\lvert\psi_2\rangle$ |
| $n^{-2}$ | $\binom{n}{2}+\binom{n}{1}$ | $\lvert\psi_1\rangle^{\otimes(n-2)}\lvert\psi_2\rangle^{\otimes 2}$ , $\lvert\psi_1\rangle^{\otimes(n-1)}\lvert\psi_3\rangle$ |
| $n^{-3}$ | $\binom{n}{3}+\binom{n-1}{1}\binom{n}{1}$ | $\lvert\psi_1\rangle^{\otimes(n-3)}\lvert\psi_2\rangle^{\otimes 3}$ , $\lvert\psi_1\rangle^{\otimes(n-2)}\lvert\psi_2\rangle\lvert\psi_3\rangle$ |
| $n^{-4}$ | $\binom{n}{4}+\binom{n-1}{2}\binom{n}{1}+\binom{n}{2}$ | $\lvert\psi_1\rangle^{\otimes(n-4)}\lvert\psi_2\rangle^{\otimes 4}$ , $\lvert\psi_1\rangle^{\otimes(n-3)}\lvert\psi_2\rangle^{\otimes 2}\lvert\psi_3\rangle$ , $\lvert\psi_1\rangle^{\otimes(n-2)}\lvert\psi_3\rangle^{\otimes 2}$ |

As shown in (Kübler et al., 2021, Theorem 2), the spectrum of a generic projected kernel has one constant (with $n$) eigenvalue, and the rest are exponentially small with $n$. Contrast that with the spectrum of the bandwidth-equipped kernel given in Table B.2. Similarly to projected kernels, in bandwidth-equipped kernels the first eigenvalue stays constant as the number of qubits grows. However, the spectrum decay behavior is different, since the eigenvalues decay polynomially and not exponentially with $n$. This leads to a qualitatively different inductive bias, which may be beneficial in some settings.

### B.4 Bandwidth and trainability of quantum neural networks

The phenomenon of flat spectrum of the quantum kernels is deeply connected to the barren plateaus phenomenon in quantum neural networks (QNNs) (McClean et al., 2018) since both stem from the exponential dimensionality of the space in which the classical data points are embedded (Kübler et al., 2021; Holmes et al., 2022). "Barren plateaus" in the context of QNNs refers to the gradients of the loss function becoming exponentially small with the number of qubits for sufficiently deep QNNs because of the loss function concentrating around its mean in high-dimensional quantum Hilbert space. Notably, a mechanism analogous to rescaling the bandwidth has been observed to enable training of quantum neural networks. Zhang et al. (2022) show that if the parameters are initialized from a Gaussian distribution with zero mean and variance $O(\frac{1}{L})$ for circuits of depth $L$, the barren plateaus are provably avoided. Rescaling the initialization of trainable parameters avoids barren plateaus by limiting the effective dimensionality of the subspace of the quantum Hilbert space being used. This is analogous to how scaling down of the data controls the bandwidth and the spectrum of quantum kernels, with the connection coming from the equivalency between quantum kernel methods and infinitely deep QNNs (Schuld, 2021). Recently, the connection between the trainability of QNNs and quantum kernel methods has been extended by Thanasilp et al. (2022) to show that several mechanisms that make QNNs impossible to train efficiently also prevent efficient training of analogous quantum kernels.

## C  Generalization Error in Kernel Ridge Regression

In this section we review the theoretical generalization error curves for kernel ridge regression developed by Bordelon et al. (2020); Canatar et al. (2021) and extend our results to the cases with a nonzero ridge parameter and noise on the labels. For kernel machines, a reproducing kernel Hilbert space $\mathcal{H}$ defines a set of functions over which the empirical loss function is minimized. Consider a training set $\mathcal{D} = \{\mathbf{x}^\mu, y^\mu\}_{\mu=1}^P$, where the inputs are drawn i.i.d. from a distribution $p(\mathbf{x})$ on $\mathbf{x} \in \mathcal{X}$ and the labels are generated through a target function $\bar{f}(\mathbf{x})$ as $y^\mu = \bar{f}(\mathbf{x}^\mu) + \epsilon^\mu$, where $\epsilon^\mu \sim \mathcal{N}(0, \sigma^2)$ is an additive noise with variance $\sigma^2$. Then the predictor is given by minimizing the empirical mean-squared-error over $\mathcal{H}$:

$$f^*(\mathbf{x}) = \arg\min_{f \in \mathcal{H}} \frac{1}{2} \sum_{\mu=1}^P \left(f(\mathbf{x}^\mu) - y^\mu\right)^2 + \frac{\lambda}{2}\|f\|_{\mathcal{H}}^2, \tag{A.11}$$

where $\lambda$ is the ridge parameter regularizing the Hilbert norm of the predictor. Associated with the RKHS $\mathcal{H}$ is a positive semi-definite kernel $k(\mathbf{x}, \mathbf{x}')$ satisfying the reproducing property:

$$\langle k(\mathbf{x}, \cdot), f(\cdot)\rangle_{\mathcal{H}} = f(\mathbf{x}),$$

where $\langle \cdot, \cdot \rangle_{\mathcal{H}}$ is the Hilbert inner product on $\mathcal{H}$. A basis for $\mathcal{H}$ can be obtained by solving the integral eigenvalue problem with respect to the input distribution $p(\mathbf{x})$:

$$\int k(\mathbf{x}, \mathbf{x}')\phi_k(\mathbf{x}')p(\mathbf{x}')d\mathbf{x}' = \eta_k \phi_k(\mathbf{x}), \quad \int \phi_k(\mathbf{x})\phi_l(\mathbf{x})p(\mathbf{x})d\mathbf{x} = \delta_{kl},$$

where $\{\phi_k(\mathbf{x})\}$ is a basis for $L_2(\mathcal{X})$ with respect to the distribution $p(\mathbf{x})$. A normalized basis for the RKHS $\mathcal{H}$ is obtained by the *features* $\psi_k(\mathbf{x}) \equiv \sqrt{\eta_k}\phi_k(\mathbf{x})$ that satisfy

$$\langle \psi_k(\cdot), \psi_l(\cdot) \rangle_{\mathcal{H}} = \delta_{kl}.$$

With these bases, the kernel can be decomposed as

$$k(\mathbf{x}, \mathbf{x}') = \sum_k \eta_k \phi_k(\mathbf{x})\phi_k(\mathbf{x}') = \sum_k \psi_k(\mathbf{x})\psi_k(\mathbf{x}').$$

Furthermore, any target function in $L_2(\mathcal{X})$ can be decomposed as

$$\bar{f}(\mathbf{x}) = \sum_k \bar{a}_k \phi_k(\mathbf{x}) = \sum_k \frac{\bar{a}_k}{\sqrt{\eta_k}}\psi_k(\mathbf{x}), \quad \|f\|_{\mathcal{H}}^2 = \sum_k \frac{\bar{a}_k^2}{\eta_k}.$$

Note that a function belongs to the RKHS only if $\|f\|_{\mathcal{H}} < \infty$. In the case where $K(\mathbf{x}, \mathbf{x}')$ has zero eigenvalues while the function $f$ has components along the corresponding eigenfunctions, this function is said to be out-of-RKHS (since $\|f\|_{\mathcal{H}} = \infty$), and a kernel machine can learn only the components along the nonzero eigenvalues. We show that the formula for generalization error in Canatar et al. (2021, Eq. 4) also extends to out-of-RKHS targets by appropriately taking the $\eta_k \to 0$ limit.

Given the $P \times P$ kernel Gram matrix $\mathbf{K}_{\mu\nu} = k(\mathbf{x}^\mu, \mathbf{x}^\nu)$, the solution to the kernel regression problem Eq. A.11 is

$$f^*(\mathbf{x}) = \mathbf{k}(\mathbf{x})^\top (\mathbf{K} + \lambda\mathbf{I})^{-1} \mathbf{y},$$

where $\mathbf{k}(\mathbf{x})$ is a $P$-dimensional vector with components $\mathbf{k}(\mathbf{x})_\mu = k(\mathbf{x}, \mathbf{x}^\mu)$ and $\mathbf{y}_\mu = y^\mu$ is a $P$-dimensional vector of the labels. Then generalization error, as function of number of training samples $P$ and the dataset $\mathcal{D}$, is defined as

$$E_g(P, \mathcal{D}) = \int \left( f^*(\mathbf{x}) - \bar{f}(\mathbf{x}) \right)^2 p(\mathbf{x})d\mathbf{x}.$$

However, the dependency of this quantity to the particular choices of datasets of size $P$ makes it analytically intractable. Instead, we are interested in the averaged generalization error $\langle E_g(P, \mathcal{D}) \rangle_{\mathcal{D}}$ over the datasets of size $P$, which has been calculated using replica theory in Canatar et al. (2021).

As a function of number of training samples $P$, ridge parameter $\lambda$, and variance of label noise $\sigma^2$ as well as the kernel eigenvalues $\{\eta_k\}$ and the target weights $\{\bar{a}_k\}$, the result for $\langle E_g(P, \mathcal{D}) \rangle_{\mathcal{D}}$ becomes (Canatar et al., 2021):

$$E_g(P) = \frac{\kappa^2}{1 - \gamma} \sum_k \frac{\bar{a}_k^2}{(P\eta_k + \kappa)^2} + \sigma^2 \frac{\gamma}{1 - \gamma}, \qquad (A.12)$$

where

$$\kappa = \lambda + \kappa \sum_k \frac{\eta_k}{P\eta_k + \kappa}, \quad \gamma = \sum_k \frac{P\eta_k^2}{(P\eta_k + \kappa)^2}.$$

Here $\kappa$ is to be solved self-consistently, and it acts as an effective ridge parameter that depends on the kernel eigenvalues and number of training samples. Even in the absence of an explicit ridge parameter (i.e. $\lambda = 0$), implicit regularization prevents the predictor from having large variance.

### C.1 Derivation of Eq. 9 in main text

In Sec. 4 of the main text and in Appendix B, we have shown that the top eigenvalues $\eta_{k,z}$ of the kernel in Eq. 6 scale polynomially with the number of input dimensions, that is, $\eta_{k,z} \sim \mathcal{O}(n^{-k})$, where the index $k = 1, 2, \ldots$ represents different scalings and index $z = 1, \ldots, N(n, k)$ represents the degenerate modes in scaling $k$. Note that the number of degenerate modes $N(n, k)$ grows as $\mathcal{O}(n^k)$ such that $\bar{\eta}_k \equiv N(n, k)\eta_k \sim \mathcal{O}(1)$. We also decomposed the target function onto the kernel basis as

$$\bar{a}_{k,z} = \int \bar{f}(\mathbf{x})\phi_{k,z}(\mathbf{x})p(\mathbf{x})d\mathbf{x}$$

and defined $\bar{a}_k^2 \equiv \sum_{z=1}^{N(n,k)} \bar{a}_{k,z}^2$ as the total weight at scaling $k$. Plugging these quantities in Eq. A.12, we get

$$E_g(P) = \frac{\kappa^2}{1 - \gamma} \sum_k \sum_{z=1}^{N(n,k)} \frac{\bar{a}_{k,z}^2}{(P\eta_{k,z} + \kappa)^2} + \sigma^2 \frac{\gamma}{1 - \gamma},$$

$$\kappa = \lambda + \kappa \sum_k \sum_{z=1}^{N(n,k)} \frac{\eta_{k,z}}{P\eta_{k,z} + \kappa}, \quad \gamma = \sum_k \sum_{z=1}^{N(n,k)} \frac{P\eta_{k,z}^2}{(P\eta_{k,z} + \kappa)^2}.$$

Furthermore, we made the approximation that $\eta_{k,z} \approx \eta_{k,z'}$ for all $z, z'$ since, in large $n$ limit, modes in the same scaling differ from each other with some $\mathcal{O}_n(1)$ quantity. Hence, we drop the index $z$. Then, in terms of $\bar{\eta}_k \equiv N(n, k)\eta_k$, generalization error simplifies to

$$E_g(P) = \frac{\kappa^2}{1 - \gamma} \sum_k \frac{\bar{a}_k^2}{(\alpha_k \bar{\eta}_k + \kappa)^2} + \sigma^2 \frac{\gamma}{1 - \gamma},$$

$$\kappa = \lambda + \kappa \sum_k \frac{\bar{\eta}_k}{\alpha_k \bar{\eta}_k + \kappa}, \quad \gamma = \sum_k \frac{\alpha_k \bar{\eta}_k^2}{(\alpha_k \bar{\eta}_k + \kappa)^2},$$

where we define $\alpha_k \equiv \frac{P}{N(n,k)}$ denoting the learning stage. Now, consider the number of samples $P$ to scale with $\mathcal{O}(n^l)$ for some integer $l$. Since $N(n, k) \sim \mathcal{O}(n^l)$, as $n \to \infty$ the quantity $\alpha_k$ becomes

$$\alpha_{k<l} = \infty,$$
$$\alpha_{k=l} \sim \mathcal{O}(1),$$
$$\alpha_{k<l} \approx 0.$$

Therefore, in the large $n$ limit, generalization error simplifies greatly and becomes

$$E_g(P) = \frac{\kappa^2}{1 - \gamma} \frac{\bar{a}_l^2}{(\alpha_l \bar{\eta}_l + \kappa)^2} + \tilde{\sigma}^2 \frac{\gamma}{1 - \gamma} + \sum_{k>l} \bar{a}_k^2, \quad \gamma = \frac{\alpha_l \bar{\eta}_l^2}{(\alpha_l \bar{\eta}_l + \kappa)^2},$$

where $\tilde{\sigma}^2 = \sigma^2 + \sum_{k>l} \bar{a}_k^2$ is the effective noise and $\kappa$ has an explicit solution:

$$\kappa = \begin{cases} \frac{1}{2}(\tilde{\lambda} + \bar{\eta}_l - \bar{\eta}_l \alpha_l)\left(1 + \sqrt{1 + \frac{4\tilde{\lambda}\bar{\eta}_l \alpha_l}{(\tilde{\lambda} + \bar{\eta}_l - \bar{\eta}_l \alpha_l)^2}}\right) & \alpha_l \leq 1 + \tilde{\lambda}/\bar{\eta}_l \\ \frac{1}{2}(\tilde{\lambda} + \bar{\eta}_l - \bar{\eta}_l \alpha_l)\left(1 - \sqrt{1 + \frac{4\tilde{\lambda}\bar{\eta}_l \alpha_l}{(\tilde{\lambda} + \bar{\eta}_l - \bar{\eta}_l \alpha_l)^2}}\right) & \alpha_l \geq 1 + \tilde{\lambda}/\bar{\eta}_l \end{cases},$$

where $\tilde{\lambda} = \lambda + \sum_{k>l} \bar{\eta}_k$ is the effective ridge parameter and describes the implicit regularization of the kernel model. Note that the total power beyond mode-$l$ acts as label noise and also irreducible error.

## C.2 Out-of-RKHS target functions and label noise

We treat the case where target function has out-of-RKHS components by setting eigenvalues with indices in an index set $\mathcal{I}$ in the generalization error formula to zero; that is, $\eta_{k \in \mathcal{I}} = 0$:

$$E_g(P) = \frac{\kappa^2}{1-\gamma} \sum_k \frac{\bar{a}_k^2}{(P\eta_k + \kappa)^2} + \sigma^2 \frac{\gamma}{1-\gamma}$$

$$= \frac{\kappa^2}{1-\gamma} \sum_{k \notin \mathcal{I}} \frac{\bar{a}_k^2}{(P\eta_k + \kappa)^2} + \left(\sigma^2 + \sum_{k \in \mathcal{I}} \bar{a}_k^2\right) \frac{\gamma}{1-\gamma} + \sum_{k \in \mathcal{I}} \bar{a}_k^2,$$

where $\kappa$ and $\gamma$ are again

$$\kappa = \lambda + \kappa \sum_{k \notin \mathcal{I}} \frac{\eta_k}{P\eta_k + \kappa}, \quad \gamma = \sum_{k \notin \mathcal{I}} \frac{P\eta_k^2}{(P\eta_k + \kappa)^2}.$$

Notice that target power placed on the modes corresponding to zero eigenvalues act both as label noise and irreducible error (Canatar et al., 2021). This implies that the inaccessible modes in target function due to small training set sizes simply lie outside the *effective* RKHS defined by the accessible, large eigenvalues. This also implies that very large bandwidths can also impair generalization; for $c \approx 0$ only a single non-zero eigenvalue survives and therefore generic targets which may have many modes corresponding to the remaining zero eigenvalues lie outside the effective RKHS. Therefore, the extreme case of very large bandwidth also creates a problem. This is just a restatement of the well-known bias-variance trade-off.

## D Bandwidth Makes Bounds on Generalization Vacuous

The following is a sketch of how the main theorem of Kübler et al. (2021) becomes vacuous ("fails") in a certain context where we can guarantee that $k(x, x')$ is lower bounded by some bandwidth-dependent constant.

We first note that this cannot be a positive proof: nothing about lower bounding $k(x, x')$ can provide a guarantee on classifier accuracy. Rather, this scenario just shows a context in which a lower bound on the ridge regression classifier accuracy based on the largest eigenvalue $\eta_{\max}$ is no longer effective at showing a failure of quantum kernel methods.

Now, we suppose that we can use bandwidth to require that states are "not too far away" in Hilbert space. More specifically, we suppose that there is some function $\Delta_c$ such that

$$k(\mathbf{x}^\mu, \mathbf{x}^\nu) \geq \Delta_c. \tag{A.13}$$

Note that this does not interfere with the requirement that $k$ is $L_2^\mu$ integrable since we assume that $k$ is defined on a restricted support $x \in [-\pi, \pi]^n$ (without this assumption, $k$ is not integrable in a more general treatment). We will show that some choice of $\Delta_c$ always ensures that the largest eigenvalue of $\mathbf{K}$ (and therefore the largest eigenvalue of $T_k$ with high probability) is bounded. Denote the eigenvalues of $\mathbf{K}$ as $\eta_k$, and in particular let $\eta_{\max}$ be the largest eigenvalue of $\mathbf{K}$ and $\mathbf{K}u = \eta_{\max}u$ for some eigenvector $u \in \mathbb{R}^P$. By definition, $\eta_{\max} = \max_{\|v\|=1}\langle v, \mathbf{K}v \rangle$ for all $v \in \mathbb{R}^P$, and in particular

$$\eta_{\max} = \frac{\langle u, \mathbf{K}u \rangle}{\langle u, u \rangle} \geq \frac{\langle \mathbf{1}, \mathbf{K1} \rangle}{\langle \mathbf{1}, \mathbf{1} \rangle} = \frac{1}{P} \sum_{\mu,\nu=1}^P \mathbf{K}_{\mu\nu} \geq (P-1)\Delta_c,$$

where $\mathbf{1}$ is the vector of all ones and we have applied the inequality of Eq. A.13. Proposition 10 of Rosasco et al. (2010) states that

$$\Pr\left(\sup_k \left|\frac{\eta_k}{P} - \gamma_k\right| \leq 2\sqrt{\frac{\log\left(4/\delta^2\right)}{P}}\right) \geq 1 - \delta,$$

where we contrast the empirical eigenvalues $\eta_k$ with the eigenvalues $\gamma_k$ of the integral operator $T_k$. Note that the empirical eigenvalues $\eta_k \to \gamma_k$ as $P \to \infty$. Hence, with probability at least $1 - \delta$ it holds that

$$\left| \eta_{\max} - \left( \frac{P-1}{P} \right) \Delta_c \right| \leq 2 \sqrt{\frac{\log(4/\delta^2)}{P}}. \tag{A.14}$$

Theorem 1 of Kübler et al. (2021) states that, with probability at least $1 - \epsilon - \eta_{\max} P^4$, the empirical risk for KRR with penalty $\lambda$ and $P$ training data is lower bounded by

$$R_{\mathrm{emp}}(f_m^\lambda) \geq \left( 1 - \sqrt{\frac{2\eta_{\max} P^2}{\epsilon}} \right) \|f\|_2. \tag{A.15}$$

From Eq. A.14, however, we see that $\eta_{\max}$ approaches a constant $\Delta_c$ at a rate of $O(1/\sqrt{P})$; for sufficiently large $P$, Eq. A.15 holds with vanishing probability, and for all other choices of $P$ the bound becomes vacuous with high probability under the condition that $\sqrt{2\eta_{\max} P^2/\epsilon} \geq 1$, which may be achieved within $O(1/\sqrt{P})$ precision by choosing

$$\Delta_c \geq \frac{P}{P-1} \sqrt{\frac{\epsilon}{2P^2}}.$$

Importantly, this outcome does *not* guarantee that KRR using $k$ satisfying the inequality in Eq. A.13 will achieve good generalization error. In particular, the choice $\Delta_c = 1$ will result in $T_k$ having a single nonzero eigenvalue associated with the constant function. Rather, this demonstration reemphasizes that there are two conditions to successful classification using quantum kernels: (i) the kernel should be chosen such that the eigenfunctions of $T_k$ align with the target function, and (ii) the corresponding eigenvalues should be large. By lower bounding $k$ in this way, we can guarantee condition (ii); however, successful generalization will still depend on a choice of $k$ satisfying condition (i).

# E    Numerical Methods

## E.1    Experiments with toy model

In Figs. 1 and 2 in the main text, we consider the toy kernel $k(\mathbf{x}, \mathbf{x}') = \prod_{i=1}^{n} \cos \left( c \frac{(\mathbf{x}_i - \mathbf{x}'_i)}{2} \right)$ for varying bandwidth parameter $c$ and $n$-dimensional input data $\mathbf{x} \in [-\pi, \pi]^n$ and drawn uniformly (Kübler et al., 2021). We generate a dataset of size $P$ by uniformly sampling $P$ input points and computing the corresponding labels using a target function $\bar{f}(\mathbf{x})$. We denote the vector of labels by $\bar{\mathbf{y}} \in \mathbb{R}^P$ and denote the kernel Gram matrix by $\mathbf{K}$ whose elements are $\mathbf{K}_{\mu\nu} = k(\mathbf{x}^\mu, \mathbf{x}^\nu)$. We obtain the eigenvalues and eigenvectors of the kernel by solving the empirical eigenvalue problem:

$$\frac{1}{P} \sum_{\nu=1}^{P} \mathbf{K}_{\mu\nu} \mathbf{\Phi}_{\nu,k} = \eta_k \mathbf{\Phi}_{\mu,k}, \quad \frac{1}{P} \sum_{\mu=1}^{P} \mathbf{\Phi}_{\mu,k} \mathbf{\Phi}_{\mu,l} = \delta_{kl},$$

where $\mathbf{\Phi}_{\mu,k}$ is the matrix of eigenvalues whose columns are the orthonormal eigenvectors and $\{\eta_k\}_{k=1}^{P}$ are the eigenvalues. Note that we obtain at most $P$ eigenmodes with $P$ samples and hence $k$ runs from $1, \ldots, P$. Finally, we obtain the target weights by projecting the targets on the eigenvectors of $\mathbf{K}$:

$$\mathbf{a} = \frac{1}{P} \mathbf{\Phi}^\top \bar{\mathbf{y}}.$$

Using the eigenvalues $\{\eta_k\}_{k=1}^{P}$ and target weights $\mathbf{a}$, we directly compute the generalization error by plugging them in Eq. A.12. To perform the experiments, we used the Kernel Generalization code by Canatar et al. (2021) and utilized a single NVIDIA V100 GPU with 32 GB of RAM. In Fig. E.1, we present the same experiment as Fig. 2 but for different input dimensions $n$. We find that the optimal bandwidth parameter is $c^* = 2/n$. Therefore, the optimal scaling of the bandwidth parameter is $\mathcal{O}(n^{-\alpha})$ for $\alpha = 1$ in this special case.

Note that bandwidth changes both the eigenvalues and the eigenfunctions of the kernel that affect spectral bias and task-model alignment, respectively. For certain tasks, faster decaying bandwidths might improve the task-model alignment and hence yield better generalization.

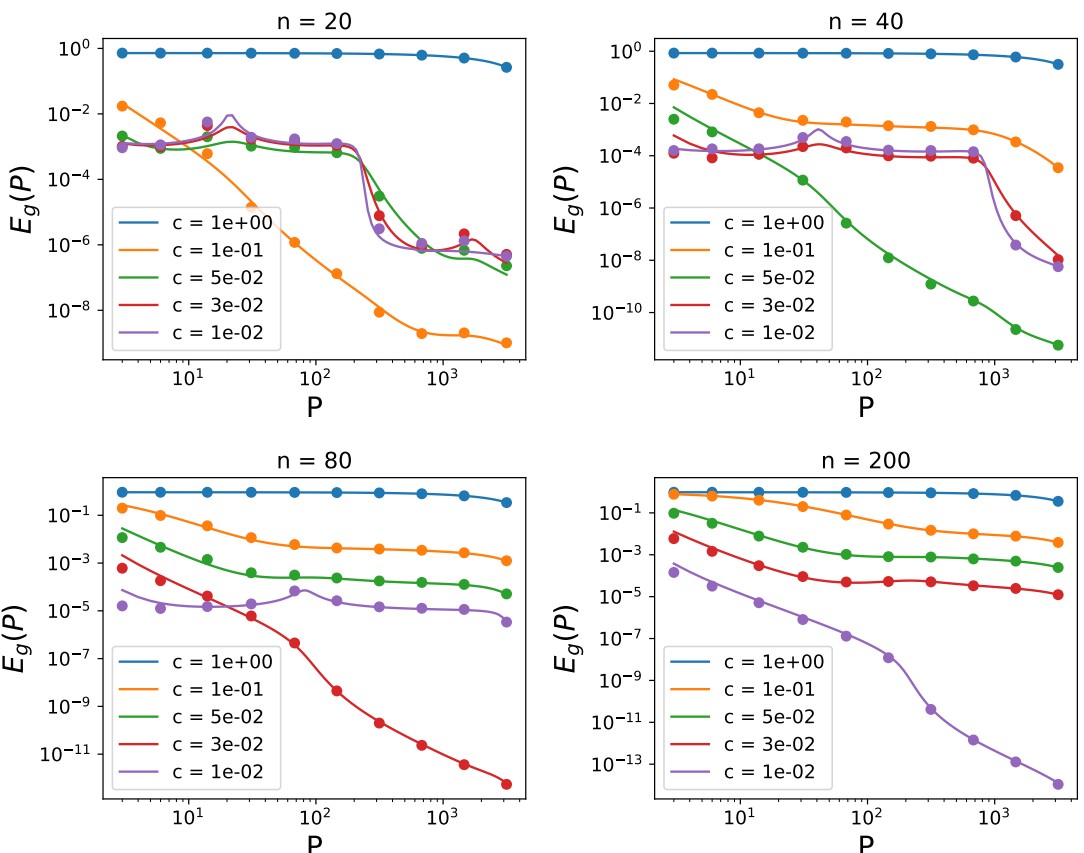

Figure A.1: Generalization error as a function of the number of training samples computed by using both theory (solid lines) and performing kernel ridge regression empirically (dots). The target function is $\bar{f}(\mathbf{x}) = e^{-\|\mathbf{x}\|^2/n^2}$, and data is drawn uniformly from $\text{Unif}([-\pi, \pi]^n)$ for $n = 20, 40, 80, 200$. Bandwidth $c = 1$ yields a constant learning curve. While all $c < 1$ kernels provide improvement, there is an optimal bandwidth parameter $c^* \approx 2/n$ that gives the best task-model alignment. Regularization with a small ridge parameter $\lambda = 10^{-10}$ is applied for numerical stability.

### E.2 Bandwidth in quantum machine learning architectures

We use the experimental data provided by Shaydulin & Wild (2021).[1] We use the kernel given by the instantaneous quantum polynomial-time (IQP) circuit feature map (Shepherd & Bremner, 2009; Havlíček et al., 2019; Huang et al., 2021) and Hamiltonian evolution circuit (EVO) feature map(Huang et al., 2021; Shaydulin & Wild, 2021) and real datasets FMNIST (Xiao et al., 2017), KMNIST (Clanuwat et al., 2018), and PLAsTiCC (The PLAsTiCC team et al., 2018).

Since the dimensionality of the datapoints in these datasets is too large (e.g., 784 for FMNIST and KMNIST) and leads to quantum circuits that cannot be simulated by using available tools, the inputs were downsized to 22-dimensions by using PCA (Shaydulin & Wild, 2021). Following Shaydulin & Wild (2021); Huang et al.

---

[1]The code and the data are publicly provided by Shaydulin & Wild (2021) at `https://github.com/rsln-s/Importance-of-Kernel-Bandwidth-in-Quantum-Machine-Learning/`.

(2021), we consider a binary classification problem where for each dataset, only two classes were chosen (see Shaydulin & Wild (2021) for the details on data preprocessing).

For $n$-dimensional inputs, the quantum circuit used to compute the $n$-qubit IQP kernel is given by

$$U_{\text{IQP}}(\mathbf{x}) = U_Z(\mathbf{x})H^{\otimes n}U_Z(\mathbf{x})H^{\otimes n}, \quad U_Z(\mathbf{x}) = \exp\left(c\sum_{j=1}^{n} x_j \text{Z}_j + c^2 \sum_{j,j'=1}^{n} x_j x_{j'} \text{Z}_j \text{Z}_{j'}\right),$$

where H is the Hadamard gate and Z is the Pauli Z-gate (see Havlíček et al. (2019)). This unitary acts on the $n$-qubit ground state to embed an input $\mathbf{x}^\mu$ to a quantum state $|\mathbf{x}^\mu\rangle = U_{\text{IQP}}(\mathbf{x}^\mu)|0\rangle^{\otimes n}$. Then the resulting feature map is given by $\rho_{\text{IQP}}(\mathbf{x}^\mu) = |\mathbf{x}^\mu\rangle\langle\mathbf{x}^\mu|$ with the corresponding quantum kernel $K_{\text{IQP}}(\mathbf{x}^\mu, \mathbf{x}^\nu) = \text{Tr}\left(\rho_{\text{IQP}}(\mathbf{x}^\mu)\rho_{\text{IQP}}(\mathbf{x}^\nu)\right)$.

For $n$-dimensional inputs, the quantum circuit used to compute the EVO kernel (Huang et al., 2021; Shaydulin & Wild, 2021) has $n+1$ qubits and is given by

$$U_{\text{EVO}}(\mathbf{x}) = \prod_{j=1}^{n} e^{-icx_{ij}(\text{X}_j\text{X}_{j+1} + \text{Y}_j\text{Y}_{j+1} + \text{Z}_j\text{Z}_{j+1})}.$$

Here, $c$ parameterizes time evolution and corresponds to the bandwidth of the resulting kernel. The initial $(n+1)$-qubit state is given by

$$|\Psi_0\rangle = \bigotimes_{j=1}^{n+1} |\psi_j\rangle,$$

where each $|\psi_j\rangle$ is randomly generated with respect to a single-qubit Haar measure (Huang et al., 2021). Then the quantum embedding of a sample $\mathbf{x}^\mu$ is $|\mathbf{x}^\mu\rangle = U_{\text{EVO}}(\mathbf{x}^\mu)|\Psi_0\rangle$ with the feature map $\rho_{\text{EVO}}(\mathbf{x}^\mu) = |\mathbf{x}^\mu\rangle\langle\mathbf{x}^\mu|$ and kernel $K_{\text{EVO}}(\mathbf{x}^\mu, \mathbf{x}^\nu) = \text{Tr}\left(\rho_{\text{EVO}}(\mathbf{x}^\mu)\rho_{\text{EVO}}(\mathbf{x}^\nu)\right)$.

In both cases, the resulting kernel is conjectured to be intractable to compute analytically, and both models utilize Hilbert spaces that are exponentially large in the number of qubits $n$. These quantum circuits were simulated in Shaydulin & Wild (2021) using Qiskit (Abraham et al., 2019) software to compute the kernel Gram matrices on the data. The kernel entries were computed exactly with high precision, i.e. with no hardware or statistical (shot) noise. Then the resulting kernels were used to perform SVM for the binary classification task.

The datasets were split into 800 training sets and 200 test sets. Each input was downsized to 22 dimensions using PCA, which leads to $2^{22}$- and $2^{23}$-dimensional Hilbert spaces for IQP and EVO circuits, respectively (Shaydulin & Wild, 2021). The code for accessing and processing the data was obtained at `https://github.com/rsln-s/Importance-of-Kernel-Bandwidth-in-Quantum-Machine-Learning/`.

**Kernel Ridge Regression with Realistic Quantum Kernels**

Apart from the classification task shown in the main text, we also use the same quantum kernels to perform kernel ridge regression on real data as shown in Figure A.2. We again find that there is an optimal bandwidth for both kernels signaled by low test loss.

**Optimal Bandwidth in Realistic Quantum Kernels**

We have obtained the optimal bandwidths through $k$-fold cross-validation for the SVM task. In Figure A.3, we report our results for each kernel method on all three datasets. Furthermore, we present the eigenvalues and target weights corresponding to both quantum model on all three dataset domains in Figure A.4 and Figure A.5, respectively. We find that the spectrum in all cases are flat without the bandwidth, and that the bandwidth improves the spectral properties in all cases. However, the task alignment with the quantum kernels depends significantly on the choice of the dataset, and it remains poor even with the bandwidth cure. This is in agreement with our arguments that bandwidth does not guarantee generalization, but only enables a model to potentially generalize if the target is suitable.

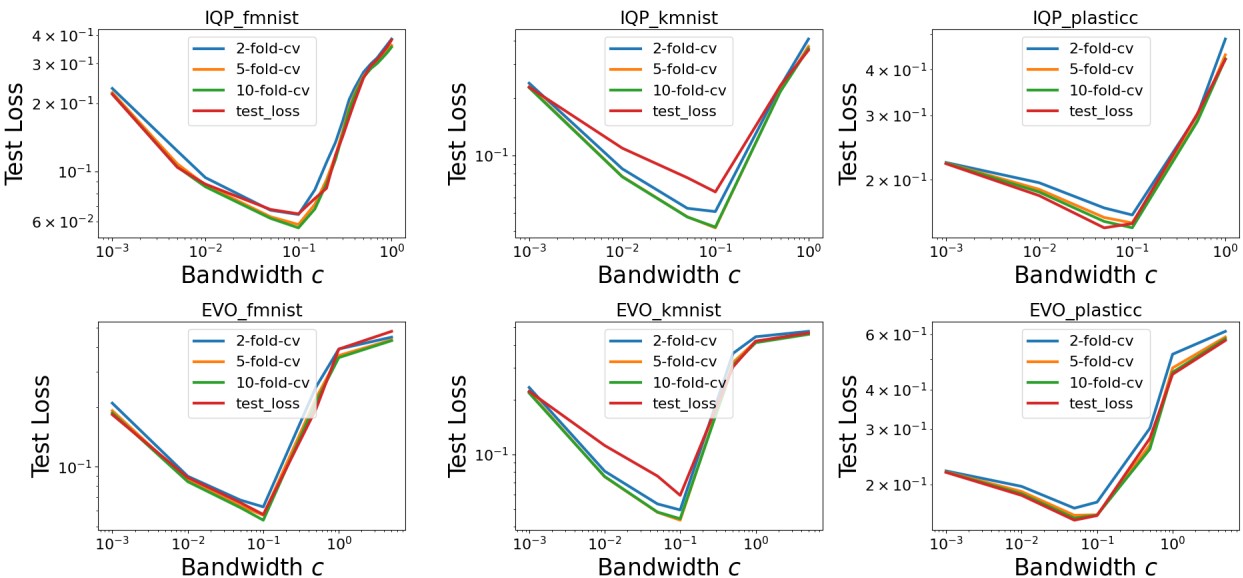

Figure A.2: Kernel Ridge Regression performed with $k$-fold cross-validation and $\lambda = 0.1$ ridge parameter yields the optimal bandwidth. The vertical axis shows the test loss of the estimator when evaluated on held-out data.

Here, we also empirically show that the optimal bandwidth scales inversely with the number of qubits $n$ as $c^* \sim \mathcal{O}(n^{-\alpha})$, where $\alpha \geq 0.5$. Using the data provided by Shaydulin & Wild (2021), we first extract the maximum kernel eigenvalue for the IQP kernel on FMNIST dataset. The IQP kernel is evaluated for various input dimensions $n$ (also the number of qubits) and various bandwidth parameters.

First, we normalize each top eigenvalue $\eta_{max}(n)$ as a function of $n$ with the maximum eigenvalue corresponding to the smallest qubit size $\eta_{max}(n_0)$, and define the quantity

$$\tilde{\eta}_{max}(n) = \frac{\eta_{max}(n)}{\eta_{max}(n_0)},$$

where $n_0 = 4$ in this case. In Figure A.6a, we plot these normalized eigenvalues against the number of qubits, and we fit exponential curves to extrapolate the behavior for large qubits which are not accessible experimentally. As the number of qubits increase, the maximum eigenvalue corresponding to $c = 1$ kernel falls much faster than the kernels with $c < 1$ bandwidth as expected. Note that in Appendix B we found that the bandwidth prevents the maximum eigenvalues to fall exponentially fast in $n$ using our toy model, and this is also what we observe here.

Next, we consider a fixed eigenvalue $\eta_0 = 0.8$ line in Figure A.6a, and in Figure A.6b we analyze where this line intersects each of the normalized eigenvalues $\tilde{\eta}_{max}(n)$ corresponding to different bandwidth parameters $c$. Similar to our calculation in Appendix B for the toy kernel, we aim to find the scaling of the bandwidth with the number of qubits such that the maximum eigenvalue stays constant. By identifying the intersection points in Figure A.6b, we obtain a relation between the optimal bandwidth and the number of qubits as shown in Figure A.6c. In this case, we numerically find that the optimal bandwidth scales as:

$$c^*(n) \propto n^{-0.506}. \tag{A.16}$$

This is in agreement with the bound for the exponent we derived in Eq. A.10. Finally, in Figure A.7 and Figure A.8, for both models when evaluated on the FMNIST data, we show the empirical scaling of the optimal bandwidth to keep their top eigenvalues constant for varying $\eta_0$'s. Again, we find that the decay exponent never violates $\alpha \geq 0.5$.

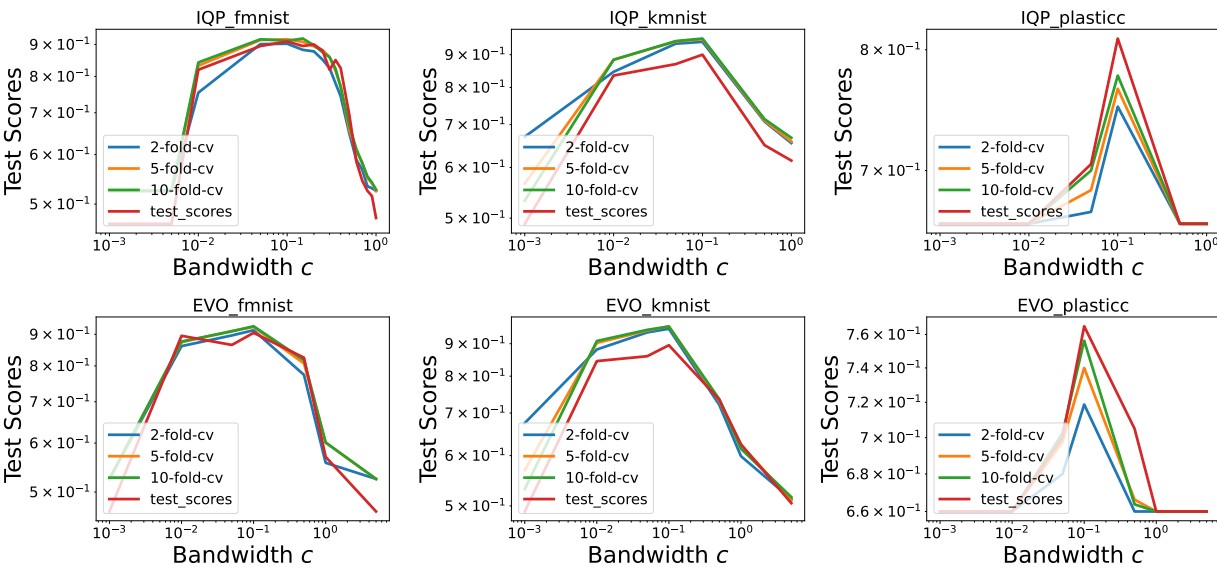

Figure A.3: SVM performed with $k$-fold cross-validation yields the optimal bandwidth.

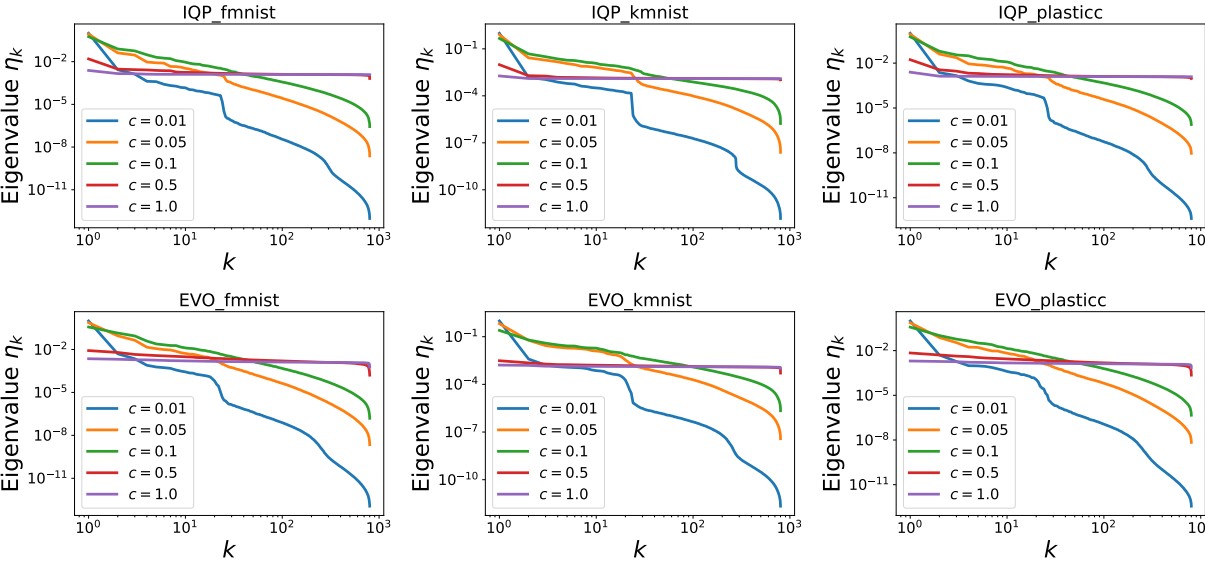

Figure A.4: Eigenvalues of the quantum kernels are always flat when the bandwidth is not tuned.

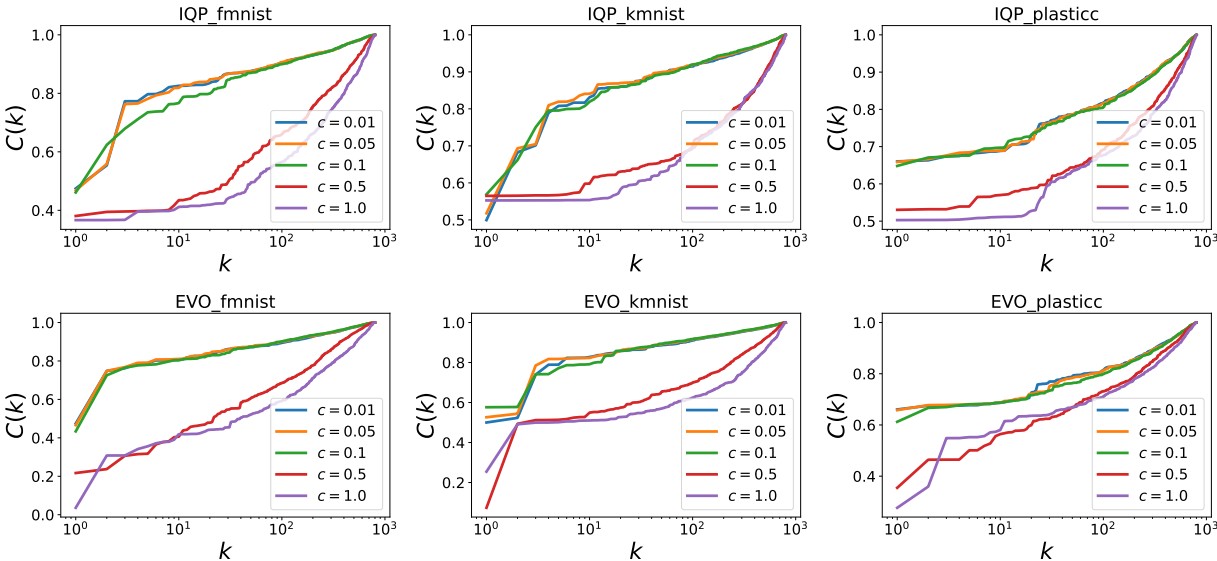

Figure A.5: Task-model alignment improves with bandwidth tuning. Note that a decaying eigenspectrum is not enough alone for generalizability. Empirically, we find that bandwidth improves both the eigenspectrum and the task-model alignment.

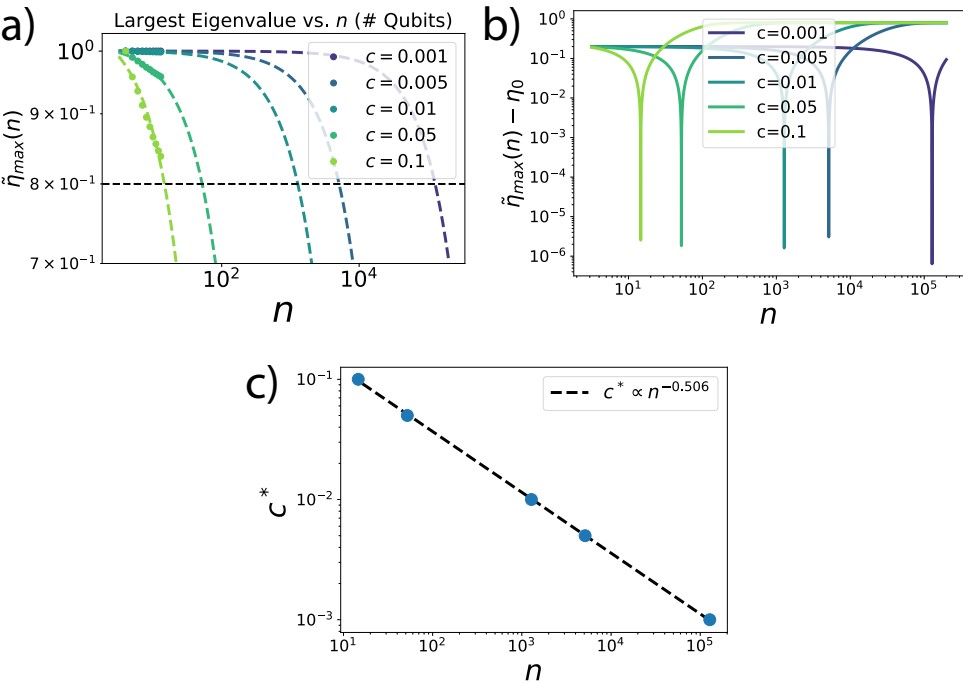

Figure A.6: Empirical scaling of the optimal bandwidth with the number of qubits.

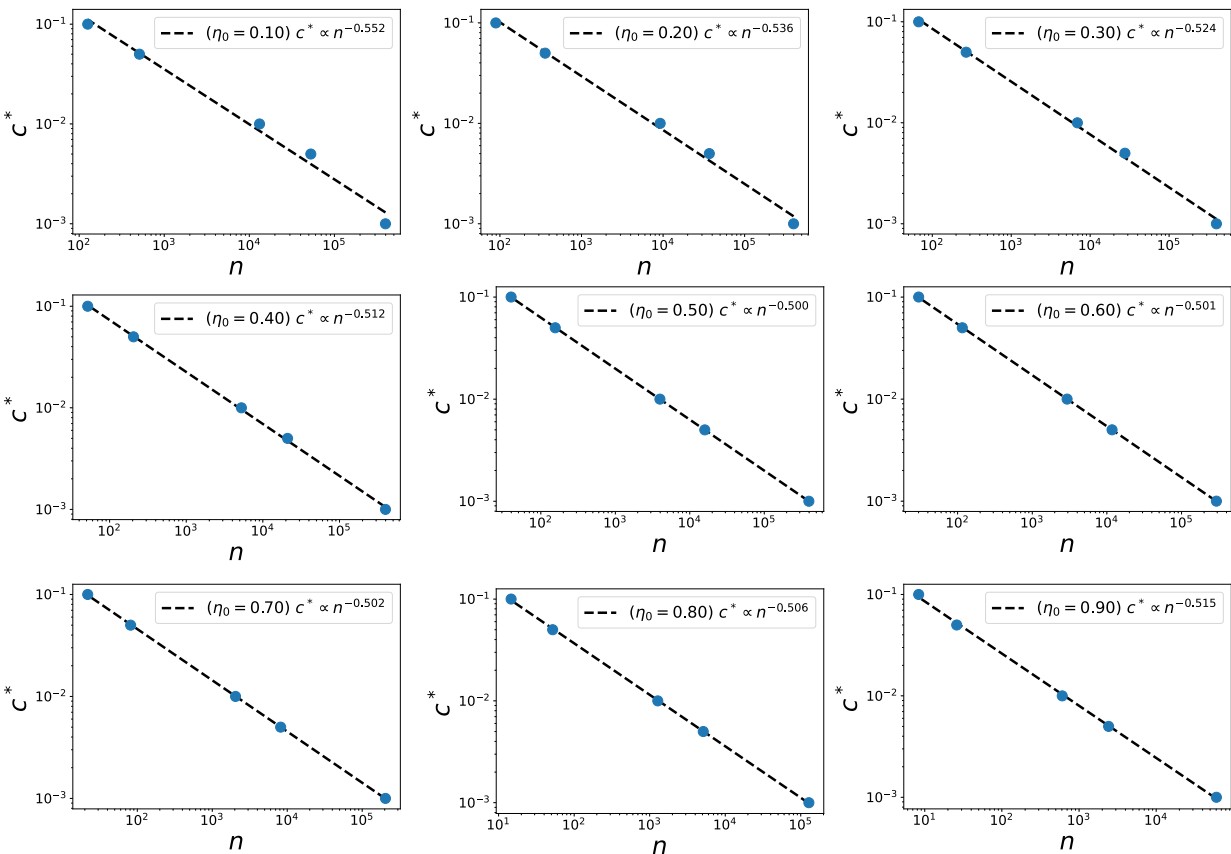

Figure A.7: IQP Kernel: Empirical scaling of the optimal bandwidth with the number of qubits for all $\eta_0$. Note that the empirical scaling never exceeds $\alpha = 0.5$ for the optimal bandwidth $c^* \approx n^{-\alpha}$.

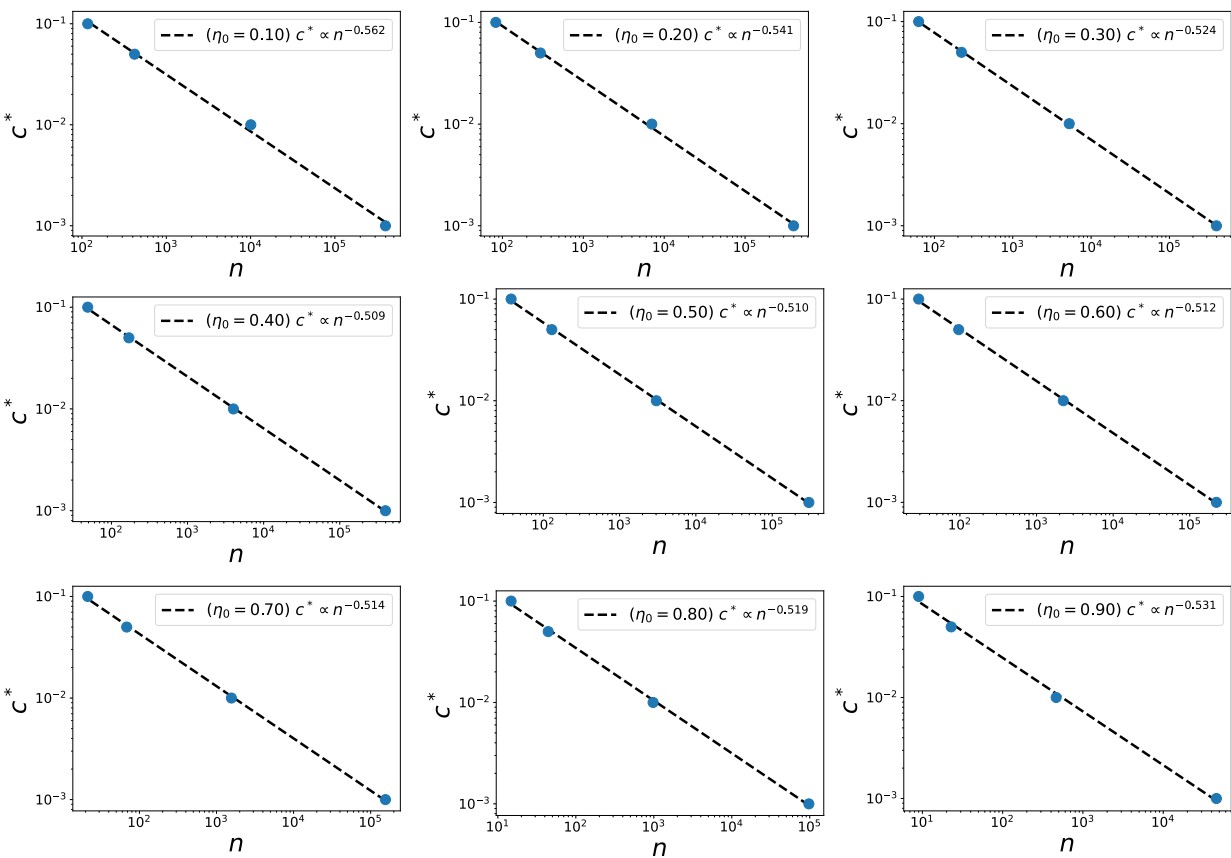

Figure A.8: EVO Kernel: Empirical scaling of the optimal bandwidth with the number of qubits for all $\eta_0$. Note that the empirical scaling never exceeds $\alpha = 0.5$ for the optimal bandwidth $c^* \approx n^{-\alpha}$.

