# OpenReview forum: "Bandwidth Enables Generalization in Quantum Kernel Models"
_TMLR — Accepted by TMLR_

### Review · Reviewer_6SfK · 2023-04-08

**Summary Of Contributions:**

The submission presents an argument for the enhanced generalization ability of quantum kernels when their bandwidth is controlled. While the authors provide a theoretical analysis of a specific Ansatz, they also conduct numerical simulations on other Ansatzes to demonstrate the practical relevance of their findings. Through experiments on real-world datasets, the authors showcase that controlling the bandwidth of many types of quantum kernels can significantly improve the performance of quantum machine learning models.

The paper is well-structured and presents some interesting results that have potential implications for the advancement of quantum machine learning. I think the paper could be published in TMLR if the issues listed in Weakness are well addressed.

**Audience:**

Yes

**Broader Impact Concerns:**

The submission targets both the machine learning and quantum machine learning communities, with potential benefits for researchers interested in designing effective quantum kernels for learning real-world datasets. However, a limitation of the paper stems from the inherent framework of quantum kernels. A recent study [Jerbi, Sofiene, et al. "Quantum machine learning beyond kernel methods." Nature Communications 14.1 (2023): 517.] has suggested that in certain scenarios, quantum kernels may be inferior to quantum neural networks. This raises questions about the necessity of using quantum kernels to tackle practical tasks.





**Claims And Evidence:**

Yes

**Requested Changes:**

The comments in Weakness should be addressed in the revised submission.

**Strengths And Weaknesses:**

Weakness:
1. The submission focuses on large-qubit quantum kernels, which may raise trainability issues, leading to the concentration phenomenon where all kernel entries become identical. This phenomenon has been discussed in a recent study by Thanasilp et al. ["Exponential concentration and untrainability in quantum kernel methods." arXiv preprint arXiv:2208.11060 (2022)]. It would be beneficial if the authors could address whether their findings are susceptible to such trainability issues and discuss any potential implications.

2. In the context of modern noisy quantum machines, the implementation of a large-qubit quantum kernel may suffer from degraded generalization ability, as demonstrated in a previous study [Wang, Xinbiao, et al. "Towards understanding the power of quantum kernels in the NISQ era." Quantum 5 (2021): 531.]. Therefore, it would be valuable for the authors to comment on whether their findings can potentially alleviate this issue.

3. The submission's theoretical analysis seems to rely on a particular ansatz, such as Eq (6). This raises two questions: Firstly, since this ansatz can be efficiently simulated by classical computers, it may not offer any significant quantum advantages. Secondly, it is unclear how the findings can be extended to other ansatzes. More discussions are expected.

---

> ### Author Response · Authors · 2023-05-11
>
> We thank the reviewer for the positive assessment of our work and a detailed review. We address the questions raised by the reviewer below.
>
> **Weaknesses**
>
> 1. We thank the reviewer for bringing reference [1] to our attention. Our work provides theoretical explanation for the mechanism observed in [2], namely that bandwidth tuning improves the generalization of quantum kernels. We note that [1] cites [2] extensively, highlighting it as one of the potential solutions to the problems raised in [1] (e.g. on page 9, second paragraph of the right column: "Thus, the key message here is that when using global measurements to evaluate the kernel, the embedding must be chosen particularly carefully<...>. To achieve this, one can <...> further reduce the expressibility of problem-agnostic embeddings [2]."). Ref. [2] finds that if the bandwidth is optimized, the quantum kernel remains trainable even when the number of qubits grows as the concentration of kernel values is avoided. The absence of concentration after bandwidth tuning is demonstrated in Figure 2d of [2]. We have added a brief discussion of Ref [1] to Appendix B4 of our manuscript.
> 2. We thank the reviewer for bringing this reference to our attention. In the last paragraph of the discussion Section of our paper, we note: "While outside the scope of this work, the impact of noise on generalization is nonetheless an important consideration for near-term prospects of quantum kernel methods." We have added the suggested reference to the discussion of the implications of modern noisy quantum machines on our results in the last paragraphs of the discussion Section.
> 3. To answer the two questions regarding the limitations of the chosen kernel and applicability of the results to a broader class of quantum kernels, we kindly point the reviewer to the last paragraph of the introduction Section, which we include here: "Our main contribution is an analysis showing explicitly the impact of quantum kernel bandwidth <..> on the generalization of the overall model. On a toy quantum model, we first demonstrate this analytically by deriving closed-form formulas <...> Furthermore, we provide numerical evidence that the same mechanism allows for successful learning for a much broader class of quantum kernels, where analytical derivation of the integral operator spectrum is impossible." In other words, we perform the theoretical analysis on the analytically tractable kernel of Eq (6) (Section 4.1), and numerically validate that our findings apply outside of our theoretical assumptions (Section 4.2). Section 4.2 provides numerical evidence and an extended discussion of the implications of analytical results to more general kernels.
>
>
> [1] Supanut Thanasilp et al. "Exponential concentration and untrainability in quantum kernel methods." arXiv:2208.11060 (2022)
> [2] Ruslan Shaydulin and Stefan M. Wild. "Importance of kernel bandwidth in quantum machine learning" arXiv:2111.05451 (2021) https://doi.org/10.1103/PhysRevA.106.042407

---

> > ### Comment · Reviewer_6SfK · 2023-06-01
> >
> > The authors have adequately addressed my concerns. I am delighted to recommend accepting this manuscript.

---

### Review · Reviewer_E8KY · 2023-04-23

**Summary Of Contributions:**

This submission investigates a variant of quantum kernel models by introducing the quantum kernel bandwidth (add a hyperparameter in the encoding or feature map). The paper follows a toy model of the quantum kernel that can be classically simulated and studies the effect of adding the hyperparameter. The paper further considers how this hyper-parameter affects the spectra and generalization error, and shows numerical experiments of the improvements of the quantum kernel model compared with the simple case.

**Audience:**

Yes

**Broader Impact Concerns:**

NA.

**Claims And Evidence:**

Yes

**Requested Changes:**

Refer to the above Strengths And Weaknesses.

**Strengths And Weaknesses:**

Strengths: The paper is well written, and the idea is interesting. The paper has done reasonable results based on their idea of quantum kernel bandwidth.

Weaknesses are summarized as follows. While the paper demonstrates some interesting aspects of quantum kernel methods, I cannot recommend acceptance for the following points:

1. Limited novelty: The effect of the bandwidth is indeed equivalent to a linear pre-processing of the data, which is a normal technique in classical machine learning. Moreover, in quantum machine learning, applying a function to the data before the encoding circuit was previously mentioned in Robust data encodings for quantum classifiers (arXiv:2003.01695). It is quite expectable that the hyperparameter could lead to different performances. The overall result does not provide any substantially novel insights or results to the quantum kernel model as well as its potential quantum advantage.

2. Limited applicability: The work mainly focuses on the tensor product quantum kernel model, which is a toy model with certain restrictions. The encoding map in this model is very costly in circuit width and can also be classically simulated. The paper needs to provide a clear and convincing explanation of how their assumptions are more general or realistic. The theoretical part needs to go deeper into the applicability and quantum advantage. This lack of clarity weakens the paper's main arguments and makes it difficult to assess the impact of their findings.

3. Not sufficient numerical simulation and its information: Although the authors claimed that "To evaluate our theory in a practical setting", the numerics are insufficient for practical purposes. For example, the information about the numerical experiments lacks clear information. Are the simulations under ideal assumption? What is the number of shots in the measurement part? Does the simulation consider any noise effects?

In summary, although the paper addresses interesting progress on quantum models, the lack of novelty, limited applicability, and insufficient numerical validation, prevent it from making a significant contribution to the field of quantum machine learning. I recommend rejection in its current form. The paper may be worth publication in a more specialized journal.

Minor comments:
- In the preliminaries, the authors introduced that a quantum state can be described by a positive definite matrix. However, it should be a positive semi-definite matrix. Similar changes should be applied to the related Hilbert space.
- The same H for feature space and quantum Hilbert space is a little bit confusing.

---

> ### Author Response · Authors · 2023-05-11
>
> We thank the reviewer for the detailed comments. Below we address the raised weaknesses individually. Based on the clarifications below, we kindly ask the reviewer to reconsider the score.
>
> **Weaknesses**
>
> 1. We thank the reviewer for raising concerns regarding the novelty of using hyperparameters to improve generalization. We are aware that bandwidth is an old and well known concept, however in this paper we argue that bandwidth scaling of inputs is required for quantum kernels simply due to its intrinsic exponential dimensionality. Note that unlike classical kernels such as Gaussian RBF that must be optimally chosen via cross validation, bandwidth does not explicitly appear as a hyperparameter for the quantum kernel. Introduction of hyperparameters is essential for the success of quantum kernel methods. Our work provides one general technique for introducing a hyperparameter into a quantum kernel. As we noted in the manuscript, there may be other techniques which can improve generalization, however bandwidth is the simplest one which had not been studied in the context of quantum kernels.
>
> 2. Our theoretical analysis relies on the toy kernel introduced in Eq. 6, while the numerical experiments presented in Section 4.2 were performed on realistic quantum kernels for well-known classification benchmarks.
>
> First, we would like to note that the central goal of this work is to explain the behavior of real quantum kernels on real data as observed in [1]. To explain these numerical results and provide a mathematical mechanism by which bandwidth enables generalization, we derive the analytical results in Section 4.1. However, in either classical or quantum settings, restrictive assumptions must be placed on the kernel to derive the analytical kernel spectrum in closed form. At the same time, studying these simple kernels provides an intuition and tests a mathematical framework for understanding the behavior of more interesting and relevant data distributions. Therefore, theoretical results in Section 4.1 use an analytically tractable kernel that is available in closed form, which is different from the more general quantum kernels studied in Section 4.2.
>
> Despite the differences in the settings chosen for theoretical and numerical analysis, the intuition developed in Section 4.1 applies to more general kernels. Specifically, we observe the same behavior (bandwidth tuning improves the spectrum and task-model alignment) for quantum kernels not available in closed form (e.g. the IQP kernel; see Figure 3) as we do for the analytically tractable kernel considered in Section 4.1. The fact that the same behavior is observed for kernels and data far outside our theoretical assumptions only reinforces that power of the developed framework.
>
> 3. We thank the reviewer for bringing up the need for additional disclosure of the numerical experiments. To answer reviewer's first question, we kindly point to the second paragraph of Section 4.2, which states: "We use the kernel values reported in Shaydulin & Wild, 2021., which were evaluated with high precision using an idealized (noiseless) simulator." To address reviewer's second question, we have extended the discussion of the experimental setup in Appendix E2 to clarify that the statement "with high precision" means that shot noise was not consider and exact values of amplitudes were used: "The kernel entries were computed exactly with high precision, i.e. with no hardware or statistical (shot) noise."
>
>
> [1] Ruslan Shaydulin and Stefan M. Wild. "Importance of kernel bandwidth in quantum machine learning" arXiv:2111.05451 (2021) https://doi.org/10.1103/PhysRevA.106.042407

---

> > ### Comment · Reviewer_E8KY · 2023-06-07
> > **Reply to updates**
> >
> > The authors have adequately addressed my concerns. I am delighted to recommend acceptance.

---

### Review · Reviewer_VUNb · 2023-05-02

**Summary Of Contributions:**

In this paper, the authors focus on the generalization properties of quantum kernel methods.
By introducing the hyper-parameter of quantum kernel bandwidth and by studying its impact on the (eigen-)spectrum of the resulting kernel integral operator, the authors show that even in the setting of large qubit, quantum kernel methods can generalize well with an appropriate choice of kernel bandwidth.

Empirical results on real-world datasets are provided to validate the theoretical results.

**Audience:**

Yes

**Broader Impact Concerns:**

The contribution of this paper is a theoretical and I do not see any ethical concerns for this paper.

**Claims And Evidence:**

Yes

**Requested Changes:**

* The term "kernel bandwidth" is reminiscent of the kernel (e.g., Gaussian) bandwidth in the literature of classical (non-quantum) kernel, it may of interest to discuss and compare how they are similar and/or different.
* Second paragraph of Section 3, starting with "One example of how high dimensionality ...": In general, I am confused by this paragraph, I do NOT really understand under which mathematical assumption the results hold (it should not be "$\phi$ consisting entirely independent Gaussian features," in which the covariance may different from identity) and I do NOT understand in which part of Appendix A is the fact "$\Sigma$ shares the eigenvalues with $T_k$" proved, and which specific result in (Rosasco et al., 2010) is cited. I would suggest that the authors should re-organize this part (and possible a few parts that follow) to state the theoretical results in a very clear manner.
* the whole section 3.1 is mainly on the example considered in (Huang et al., 2021). I am wondering if something can be said beyond this example. The same holds for Section 4.1.


**Strengths And Weaknesses:**

**Strengths**: this paper focuses on the important and timely problem of quantum kernel method and discusses its generalization performance. The idea of kernel bandwidth is of interest.

**Weaknesses**: While the core idea of this paper is not difficult to grasp, the presentation of this paper (examples instead of formal theorem, etc) makes it difficult for readers to go deeper, and in particular, makes one feel that this paper only studies special cases. See my detailed comments below.

---

> ### Author Response · Authors · 2023-05-11
>
> We thank the reviewer for the positive assessment of our results and detailed comments. We address the changes requested by the reviewer below.
>
> **Requested Changes**
>
> 1. We thank the reviewer for bringing up the connection between the quantum kernel bandwidth that we study and the bandwidth hyperprameter in classical Gaussian kernel. This connection is explored in depth in Section "Impact of bandwidth on model generalization and comparison with classical kernels" of Ref. [1]. We updated Section 5 to highlight the connection that Ref. [1] makes between classical and quantum kernel bandwidth.
> 2. We note to the reviewer that as stated in the paragraph in question, the covariance matrix of independent random variables is proportional to identity by the definition of covariance. The fact that the spectrum of the kernel matrix approaches that of the integral kernel operator as the number of datapoint grows is well-known and is not re-proven in our paper. Instead, these results are simply stated in S1-S3. We have specified in the first paragraph of Appendix A1 that the particular result that we use from (Rosasco et al., 2010) is Proposition 9, which shows that the spectrum of K concentrates around the spectrum of $\Sigma$ (using our notation).
> 3. As we state in the introduction, we first derive the analytical results on a simple kernel in Section 4.1 and then validate them numerically ourside of theoretical assumptions in Section 4.2. As the title of the Section 3 states, it simply acts as a motivating example for Section 4. The numerical results in Section 4.2 go beyond the examples of 3.1 and 4.1.
>
> [1] Ruslan Shaydulin and Stefan M. Wild. "Importance of kernel bandwidth in quantum machine learning" arXiv:2111.05451 (2021) https://doi.org/10.1103/PhysRevA.106.042407

---

> > ### Author Response · Authors · 2023-06-16
> >
> > Dear Reviewer VUNb,
> >
> > We hope our responses cleared some of your questions and concerns. We hope to hear your thoughts and/or questions about the clarifications made above.
> >
> > Best, Authors.

---

> > > ### Comment · Reviewer_VUNb · 2023-06-17
> > >
> > > I thank the authors for their time and efforts in revising the manuscript. I still would like to clarify that the statement “the covariance matrix of independent random variables is proportional to identity by the definition of covariance” is not very mathematically precise, this is true for standard Gaussian features with zero mean and (same, e.g., unit) variance, otherwise the covariance matrix is only diagonal but not necessarily proportional to identity.
> > > Otherwise, the authors have addressed my concerns. And I am happy to recommend it for publication in TMLR.

---

### Review · Reviewer_wb6i · 2023-05-06

**Summary Of Contributions:**

This paper studies generalization properties of quantum kernel methods. In particular, this paper considers the kernel method with a trigonometric kernel as in Eq. (3), and studied the generalization property of such a kernel in terms of a parameter c, named as the bandwidth of the kernel. It is observed that the smaller c is, the better the generalization is.

**Audience:**

Yes

**Broader Impact Concerns:**

Not applied in general. Nevertheless, it would be helpful if the authors can briefly discuss about the impact to near-term quantum computing devices (see also my comment above).

**Claims And Evidence:**

Yes

**Requested Changes:**

It would be helpful if the authors can contribute to my comments above related to both theory and practice sides.

In addition, I notice the following two minor points:

- In intro, the authors may also briefly discuss about Li et al. in ICML 2019, https://arxiv.org/abs/1904.02276, which studied quantum algorithms for supervised learning and also used kernel methods, in particular kernel-based classification.

- Typo: title of Appendix A: Review of Classical and Quantum Data operators -> Review of Classical and Quantum Data Operators


**Strengths And Weaknesses:**

From my perspective, the main advantage of this work is that it provides a framework for studying generalization properties of quantum machine learning algorithms with theoretical guarantee. Relevant such works are sparse as far as I know, and it’s nice to see a solid contribution along this line.

In terms of weaknesses, I think the paper can be further improved from the following two perspectives.

In theory, it would be helpful to discuss how general the kernel in Eq. (6) is. In particular, does the result also apply to other kinds of kernels, and will there be a similar bandwidth parameter c for characterizing generalization?

In practice, it would be helpful to explain how the theoretical results in this paper can be helpful for near-term quantum devices. For instance, does the result in this paper further imply generalization guarantees for VQE or QAOA algorithms?

---

> ### Author Response · Authors · 2023-05-11
>
> We thank the reviewer for the positive assesment of our work and thoughtful comments that helped improve the presentation. Below we respond to the comments point-by-point.
>
> **Weaknesses**
>
> 1. While our theoretical results focus on the simple analytically tractable kernel given in Eq. (6), we numerically validate the findings in Sec. 4.2 in a more general setting on kernels that are conjectured to be classically hard to simulate. For these kernels, closed-form spectrum is not available (the availability of such results would likely imply classical simulatability of the kernels). However, we numerically observe that a similar bandwidth parameter c improves the generalization by controlling the spectrum analogously to the theoretical results in Sec. 4.1
> 2. The analysis of the impact of noise on the near-term devices is outside of the scope of the current work. We briefly discuss the implications of our results to kernels evaluated on noisy quantum computers in the last paragraph of Sec. 6.
>
> **Requested Changes**
>
> 1. We thank the reviewer for pointing out this reference. We have added a brief discussion of it to the first paragraph of Introduction, as suggested.
> 2. We corrected the typo. Thank you for pointing it out!

---

> > ### Comment · Reviewer_wb6i · 2023-06-03
> > **Thanks for the updates**
> >
> > The authors have adequately addressed my concerns. I'm happy to recommend acceptance.

---

### Comment · Action_Editors · 2023-05-02
**Rolling discussion starts**

Dear authors,

We have collected three comments. The rolling discussion automatically starts once three comments have been collected. You will have two weeks to do rebuttals with reviewers. The system will remind the reviewers to submit their final recommendation in two weeks. Note that one more reviewer would submit a fourth review in the coming days. If you would extend the rolling discussion phase, please let me know within two weeks.

Best wishes,
Tongliang

---

### Author Response · Authors · 2023-05-11

We thank the reviewers for the detailed feedback that helped improve the presentation. We uploaded a revised manuscript. The changes to the manuscript are indicated in blue font. In comments below we address each reviewer individually.

---

### Decision · Action_Editors · 2023-06-16

**Recommendation:** Accept as is

**Comment:**

The paper explores the generalization properties of quantum kernel methods. It introduces that the hyperparameter "quantum kernel bandwidth" is important for improving the generalization of a quantum model. After rebuttal, all reviewers had a positive view of the work and  recommended acceptance, acknowledging its theoretical rigor and the relevance of the topic.

These reviewers noted the authors' theoretical analysis and empirical validation of the impact of the quantum kernel bandwidth on generalization. Reviewer wb6i also highlighted that this work is particularly significant due to a lack of existing literature on theoretical guarantees for the generalization properties of quantum machine learning algorithms. Reviewer 6SfK further recommended the Featured Certification. In light of these recommendations and discussions, I support the acceptance of this paper for publication in TMLR. Their work may open up new avenues for research and practical applications in quantum machine learning.




**Audience:**

The findings of this paper would be of interest to the audience of TMLR, especially those who work at the intersection of machine learning and quantum computing. Quantum machine learning is a burgeoning field, and there is a growing interest in understanding the theoretical underpinnings of quantum algorithms and their generalization properties. The paper presents new insights into the generalization of quantum kernel methods, which can be an important topic for researchers working on quantum machine learning.


**Claims And Evidence:**

The claims made in the submission are supported by both theoretical analysis and empirical evidence. The authors focus on the generalization properties of quantum kernel methods and propose a bandwidth parameter to improve generalization.  The authors back up their claims with theoretical analysis, providing explicit formulas for the generalization of a quantum model that can be solved in closed form. They also demonstrate empirically that their theory can predict how varying the bandwidth affects generalization on challenging datasets.

However, as highlighted by Reviewer VUNb, there is a need for mathematical precision in a certain statement made in the paper, particularly the statement “the covariance matrix of independent random variables is proportional to identity by the definition of covariance.” While the authors addressed most of the concerns raised, they will need to revise this statement for clarity and accuracy in the final version of the paper.

Despite this minor issue, the evidence presented supports the claims made, and the reviewers accepted the validity and importance of the findings. The authors' responses to the reviewers' comments further confirm the robustness of their results. Hence, the paper appears to meet the standard of rigor expected for TMLR.